# Chitosan–Silica Composites for Adsorption Application in the Treatment of Water and Wastewater from Anionic Dyes

**DOI:** 10.3390/ijms241411818

**Published:** 2023-07-23

**Authors:** Magdalena Blachnio, Malgorzata Zienkiewicz-Strzalka, Anna Derylo-Marczewska, Liudmyla V. Nosach, Eugeny F. Voronin

**Affiliations:** 1Faculty of Chemistry, Maria Curie-Sklodowska University, M. Curie-Sklodowska Sq. 3, 20-031 Lublin, Poland; malgorzata.zienkiewicz-strzalka@mail.umcs.pl (M.Z.-S.); anna.derylo-marczewska@mail.umcs.pl (A.D.-M.); 2Chuiko Institute of Surface Chemistry, National Academy of Sciences of Ukraine, 17 General Naumov Str., 03164 Kyiv, Ukraine; nosachlv@ukr.net (L.V.N.); e.voronin@bigmir.net (E.F.V.)

**Keywords:** adsorption kinetics, adsorption equilibrium, dye adsorption, chitosan–silica composite, biopolymer, chitosan, silica

## Abstract

A series of new types of composites (biopolymer–silica materials) are proposed as selective and effective adsorbents. A new procedure for the synthesis of chitosan–nanosilica composites (ChNS) and chitosan–silica gel composites (ChSG) using geometrical modification of silica and mechanosorption of chitosan is applied. The highest adsorption efficiency was achieved at pH = 2, hence the desirability of modifications aimed at stabilizing chitosan in such conditions. The amount of chitosan in the synthesis grew to 1.8 times the adsorption capacity for the nanosilica-supported materials and 1.6 times for the silica gel-based composites. The adsorption kinetics of anionic dyes (acid red AR88) was faster for ChNS than for ChSG, which results from a silica-type effect. The various structural, textural, and physicochemical aspects of the chitosan–silica adsorbents were analyzed via small-angle X-ray scattering, scanning electron microscopy, low-temperature gas (nitrogen) adsorption, and potentiometric titration, as well as their adsorption effectiveness towards selected dyes. This indicates the synergistic effect of the presence of dye-binding groups of the chitosan component, and the developed interfacial surface of the silica component in composites.

## 1. Introduction

Composite materials, which are a combination of components of sometimes very different properties into one common system, are one of the most interesting pursuits of contemporary material chemistry. Polysaccharide nanocomposites have become important materials over the last decade [1,2,3,4,5,6]. They are a novel method of preparing of soft and hard nanomaterials based on hydrogels and metal or carbon nanostructures, respectively, as well as creating a great alternative to nanocomposites based on synthetic polymers. This approach is also a response to the search for new types of composites and materials better adapted to new technologies. Aspects of their preparation remain a topical issue in the field of new types of nanocomposites, and actual challenges include obtaining desirable or unique properties, controlling their fabrication procedures, and ensuring the compatibility of their different components [7,8,9]. Polysaccharides perform many important functions in nature, from acting as cell-building materials to storing cellular energy, which are performed as a result of internal metabolic cycles [10,11,12]. As a result of the differences in the structure of the units as well as the branching of the polymer chain, they differ in their physicochemical properties. Intramolecular and intermolecular interactions between hydrogen bonds as well as the number and type of functional groups in the polysaccharide structure have a significant impact on preparing new types of nanocomposites for biomedicine, energy and electrical devices, separation science, and industrial and material applications [13,14,15,16,17,18]. The biodegradability, biocompatibility, and non-toxicity features of polysaccharides make them an attractive and viable alternative to synthetic polymers [19]. Especially important are their applications in the pharmaceutical and cosmetics industries, and in the broadly understood field of medical usage [20,21]. In the latter, the available and repeatable processing of a selected biopolymer from raw materials into a usable form is particularly important. At the moment, the possibilities of polysaccharide processing depend primarily on their solubility in different media, and their final stability. Naturally occurring polysaccharides exhibit hydrophilic properties, which, combined with the aforementioned limited solubility, result in their excellent ability to form hydrogels [22,23,24,25,26]. Over the years, among a wide group of biopolymers, chitosan has found a special place, especially in modern scientific and technological approaches. The importance of chitosan can be illustrated by the fact that it is expected to rank second after cellulose in the group of mass-produced natural polymers [27,28]. Particular attention is paid to its biological properties and applications in medicine, industry, food, and agriculture [29,30,31,32,33,34,35]. Chitosan biopolymers contain primary amine functional groups responsible for the effective adsorption of several active substances. Some valuable properties are their biocompatibility [36,37], non-toxicity, stability, and hydrophilicity. These all make chitosan a convenient carrier for the stabilization and immobilization of biologically active substances and nanophases. The incompleteness of our knowledge about multiphase biopolymer systems has forced researchers to learn and describe new properties of nanocomposites, and to specify the relationships between the main elements and their applicability.

Biopolymer composites with advanced structural and surface properties have also found an important place in adsorption processes. Among the research conducted with the use of materials, composites, and chitosan derivatives, two main routes can be distinguished. Firstly, the environmental aspect, regarding the elimination of unnecessary and undesirable substances, and secondly, the medical direction, involving the release or delivery of bioactive substances. In the first case, chitosan-based adsorbents to remove heavy metals and dyes are used due to the different types of surface active sites [38,39,40,41]. There is a great interest in the area of adsorption of toxic and undesirable compounds in water and wastewater treatment processes [42,43,44,45,46], which may also proceed through metal complexation mechanisms [47]. Otherwise, arsenicals [48] and anti-inflammatory drugs such as ibuprofen, diclofenac, and naproxen can be removed from aqueous solutions using chitosan-modified rubber composites [49]. Heavy metal pollution is a major environmental problem in the world, especially in developing countries. Thus, chitosan-based biosorbents have been tested for the biosorption of heavy metals and even radionuclides [50]. Chitosan–polyethyleneimine composites have been tested for the efficient removal of uranium from water environments [51]. Other examples have applied chitosan resin modified with L-lysine to the adsorption of metals such as platinum(IV), palladium(II), and gold(III) from aqueous solutions [52].

A widely applied issue related to chitosan is the adsorption of drugs and pharmaceuticals. This research was concerned with the study of the adsorption of propranolol hydrochloride on the surface of chitosan and cellulose acetate polymers [53], the adsorption of human serum album by chitosan microspheres [54], the adsorption and release of doxorubicin by chitosan-decorated graphene [55], tetracycline [56], or polypeptide drug luteinizing hormone [57]. Despite such versatile examples, applications related to the use of chitosan are still subject to certain limitations. In many applications, the amplification and modification of complex systems is required. The solution to this issue may be related to the creation of composite systems, e.g., silica or carbon materials [58,59,60,61].

The synthesis of organic–inorganic composites is a successful way to obtain materials with better properties in comparison to single components. The combination of many preferable features in one material presents new possibilities for potential applications. Especially promising seems to be the combination of chitosan with silica as components for the development of new hybrid adsorbents. The proposed synthesis of such adsorbents works in accordance with the requirements of the concept of “green chemistry”. The choice of chitosan and silica as synthesis reagents is closely related to limiting our use of harmful substances. It should be emphasized that these components are non-toxic, and that chitosan, as a natural polymer, is biodegradable.

It has been assumed that a chitosan–silica composite would work more effectively as an adsorbent due to the presence of the chitosan component. It is well known that due to its polycationic nature, chitosan is characterized by remarkable sorption properties towards anionic pollutants. However, it is a substance with low mechanical, thermal, and chemical stability. Therefore, the combination of chitosan with silica (whose role is to be a rigid support for the polymer due to its high stability) aims to produce a composite that would exhibit both satisfactory sorption properties and mechanical and thermal stability. To improve the chemical stability of the solid, during synthesis, it was subjected to a cross-linking process with glutaraldehyde (GA). The choice of silica as inorganic matter was also dictated by the presence of the silanol groups (Si–OH) on its surface, enabling the immobilization of organic matter. The applied procedure enabled the usage of a composite for removing anionic pollutants from aqueous solutions in various conditions (pH and temperature), and also allowed us to indicate the optimal conditions at which the adsorbent worked with the greatest efficiency.

The synthesis of the chitosan–silica composite was carried out in a ball mill equipped with ceramic balls, and the process comprised geometric modification of silica accompa-nied by chitosan mechanosorption. The applied method of synthesis aimed to provide specific conditions in the working ball mill (i.e., friction force, elevated temperature, liquid–gas phase) to evenly distribute a biopolymer on a silica surface, and to generate intermolecular interactions between them. However, the degree of homogeneity of the biopolymer film on the supporting silica surface depended on the type of silica used. This issue was investigated based on the structural and morphological results for composites synthesized from nanosilica (NS) and silica gel (SG). In the first case, due to the small size of the nanosilica particles and their relatively narrow distribution, a more homogeneous coverage was obtained with a thin layer of the organic component.

In this work, the structural modifications of chitosan–silica materials and their impact on the efficiency of the adsorption process of anionic coloring substances is examined. As a model adsorbate, Acid red 88 (AR 88), a sulfonic azo substance, was chosen. Due to the widespread usage of this group of dyes in the textile, cellulose, paper, chemical, food, and cosmetic industries, they are a main component of industrial wastewater and a dangerous source of environmental pollution. On the other hand, the complex structure of the dyes’ molecules makes them highly stable to light and oxidizing agents, as well as resistant to biodegradation. Thus, the usefulness of some traditional methods in removing dyes from aqueous solutions is limited. The effectiveness of flocculation and coagulation processes using metal compounds in water decoloration is high, but the consequence of their usage is the formation of sludge and an increase in the concentration of metallic pollutants in water. Therefore, adsorption techniques are a good alternative to conventional dye removal methods.

A comprehensive approach to adsorption process in terms of kinetics and equilibrium can provide valuable information about the adsorption mechanism of such an important biopolymer carrier. The adsorption experiment was carried out over wide pH and temperature ranges, thereby confirming the stability of the synthesized materials. It was shown that the structural characteristics of the hybrid adsorbents affect their adsorption capacity for pollutants as well as their adsorption kinetics.

## 2. Results and Discussion

### 2.1. Characterization of the Adsorbents

The investigated composites were characterized using several techniques with regard to finding the correlations between their properties and adsorption effectiveness. The functionality of the biopolymer component and the physicochemical properties of the silica phase (porosity, hydrothermal stability, and ease of surface modification) were taken into account. The names of the studied materials and their description are as follows: ChNS_05, a composite based on NS and coated with a half surface layer of chitosan; ChNS_1, a composite based on NS and coated with one surface layer of chitosan; ChNS_1_GA, a composite based on NS, coated with one surface layer of chitosan and crosslinked with glutaraldehyde; ChSG_05, a composite based on SG and coated with a half surface layer of chitosan; ChSG_1, a composite based on SG and coated with one surface layer of chitosan; and ChSG_1_GA, a composite based on SG, coated with one surface layer of chitosan and crosslinked with glutaraldehyde.

Small angle scattering (SAXS) curves for the investigated composites are presented in Figure 1A and Figure 1B, respectively. Due to the similar nature of the dispersion curves, a group of composites based on a given type of silica were analyzed together. All materials are capable of X-ray scattering in a range of small diffraction angles, i.e., they contain nanoobjects actively involved in such a phenomenon. They also show a polydispersive nature, with a lack of any structural ordering. The scattering intensities are significant, although the course of the experimental curves is very similar. In the case of nanosilica-based composites, a subtle differentiation of scattering intensity, especially in the initial range of the curves, can be identified. For the composites based on silica gel, all systems show almost identical scattering curves. This suggests that the scattering data are obtained almost entirely from the silica support, indicating the weak bonding of this support with the biopolymer component.

In the context of adsorption applications, it is necessary to determine the nature of the porous structure, including open and closed porosity. For the samples obtained with different chitosan amounts, which were introduced as an initial or cross-linked form, the parameters of the porous structure were calculated based on SAXS data (the entire porosity, including open and closed pores, which is a reflection of the differences in the electron density of the pores and the bulk material), and low-temperature nitrogen adsorption/desorption isotherms (which in turn take into account the open porosity available to the nitrogen molecules). The porosity of the silica–biopolymer composites is treated in the SAXS studies as the interfacial area between the matrix and the filler phase. To be precise, porous composites are considered hetero-phase systems in which at least two phases have a different electron density (composed of pores within a matrix phase). The small-angle X-ray scattering data were used to determine the specific surface area of the nanoporous systems, which include mesoporous and partly macroporous materials (the share of micropores is very small).

Figure 2 shows the Porod plots of experimental scattering data as the constant tail-end of the SAXS curves. The presented functions q^3^·I(q) achieve constant values in the larger range of curves. The asymptotic behavior of the SAXS curve, according to Porod’s law, is not affected by the shape and size of particles, but rather depends on the total interfacial area [62]. The difference in the level of the final region of the Porod curve determines the interfacial surface relationship. Thus, a trend of changing the interfacial area of the studied composites rather than the absolute value can be observed. For the nanocomposites based on nanoporous silica (NS), the largest interfacial area should be expected for the sample ChNS_05, a slightly lower one for ChNS_1_GA, and the lowest for ChNS_1. For the composites based on silica gel (SG), half of the chitosan layer (ChSG_05) allows us to obtain material with the highest relative specific surface area. By increasing the amount of chitosan on one surface layer (ChSG_1), we were able to reduce the interfacial surface. The cross-linking process creates a more compact structure with lower phase differentiation (ChSG_1_GA). The direct indicators here are the Porod K_P_ constants, which are summarized in Table 1 and the insets of Figure 2. Additionally, low-temperature adsorption/desorption isotherms of nitrogen for all investigated samples and correlated BJH pore size distributions from desorption branch of isotherms are presented in Appendix A.

The value of the specific surface of all investigated systems obtained from SAXS and gas physisorption do not differ significantly. In most cases, the specific surface area determined from SAXS shows slightly higher values than values determined by the physical sorption method. This trend is understandable, because scattering method provides access to all types of pores, including closed ones, which are not accessible to nitrogen molecules. The importance of closed porosity is not very high in the investigated systems, and concerns values from several to several dozen m^2^/g. Due to the potential adsorption application of the analyzed composites, the samples modified by half of the biopolymer monolayer deserve attention (ChNS_05 and ChSG_05). Here, the value of the specific surface area based on both methods is the highest, and equals 277 m^2^/g (SAXS)/244 m^2^/g (N_2_) and 269 m^2^/g (SAXS)/257 m^2^/g (N_2_) for ChNS_05 and ChSG_05, respectively. The incomplete biopolymer layer does not limit access to the pores of the silica material. In the case of introducing larger amounts of the chitosan phase, a reduction in this availability was observed, which was immediately visible in the decreasing specific surface area for such samples (ChNS_1 and ChSG_1).

It was observed that the SAXS curves were very similar for samples with different contents of chitosan, as well as for the cross-linked samples. Such a similar course itself may be a result of the essence of the SAXS method, which allows us to obtain structural information from the entire volume of the sample, and does not contribute much to the overall structural information. In other words, the SAXS data come from the silica carrier. Despite this, interesting information is provided by the curves of the volume distribution of inhomogeneity. As shown in Figure 3, the volume size distributions of both scattering groups without cross-linking demonstrate a multimodal character; several fractions of small and large scattering inhomogeneity were observed as Dv(R) function maxima with various intensities. For the materials based on nanosilica (ChNS_05, ChNS_1), a significant compartment of heterogeneity with a wide size range from 10 to 40 Å was observed. For both of these samples, the general course of the Dv(R) curve was almost identical. The cross-linking process changes the character of the curve, showing the volume distribution of heterogeneity. A shift of the maximum of the volume size distribution function of the scattering heterogeneities (Dv®) towards inhomogeneity of larger sizes was observed. The ChNS_1_GA sample was then characterized by a much narrower size distribution, containing inhomogeneity with sizes up to 60 Å.

In addition, a change in the character of the curve to a much more homogeneous course and a narrower range of visible inhomogeneity were noticed. Similar observations were made for silica gel-based composites (ChSG_05 and ChSG_1). The samples modified with a larger amount of chitosan, regardless of the amount introduced, generated a bimodal size distribution curve Dv(R) (extremum at ~10 Å and 30 Å) in a range up to 50 Å, as well as less significant variations in about 60 and 100 Å.

After cross-linking, the dispersion of inhomogeneity flattened significantly and showed only one general fraction at 20 Å. The results show that the cross-linking process significantly affects the structural order of the composite. The direct impact of this phenomenon on adsorption capacity can be assessed later in the manuscript.

In the search for an explanation of the phenomenon of such morphological changes, i.e., surface smoothing or a reduction of heterogeneity in the structure of the tested composites before and after the cross-linking process, and depending on the thickness of the biopolymer layer, a microscopic analysis was applied. Scanning electron microscopy (SEM) cross-sectional images revealed the structure of the surface and inner layers of the materials (Figure 4). The micrometer grains of the tested materials are quite compact, but at higher magnifications, the morphological features of the material phases are differentiated. The first four photos present a composite based on the same type of silica phase (nanoporous silica). Morphologically, such silica consists of spherical nanosilica particles with sizes less than 50 nm. Individual spherical forms of silica nanospheres were revealed at higher magnifications (Figure 4B,D). After modification with glutaraldehyde (GA) and the cross-linking of the biopolymer phase, the spherical morphology of the nanosilica particles does not change significantly; however, it seems that they gain an additional layer that softens the surface irregularities. The surface seems to be smoother, although the chitosan phase itself was not identified as a separated fraction. This proves the success of the composite’s synthesis from individual components. The morphology of the sample based on silica gel is different (Figure 4E–H).

The spherical nature of the material was not observed, and the sample itself had the character of an amorphous formation with no visible separations of another phase. SEM images indicated a more porous nature of the sample with a granular and spatial form. SEM analysis confirmed the SAXS observations of a more homogeneous distribution of heterogeneities. They are associated with a biopolymer layer, which forms a fairly compact layer on the material surface.

To obtain information about the chitosan content in the composites, an elemental analysis of carbon, hydrogen, and nitrogen was performed. The results are summarized in Table 2. A clear differentiation of the nitrogen content in the final material (which is related to chitosan) from the amount of added precursor was found. In the case of materials with half of the biopolymer monolayer, the mass content of nitrogen was 0.28% and 0.29% for the ChNS_05 and ChSG_05 samples, respectively. The amount almost doubled after the introduction of an amount of chitosan corresponding to the full surface monolayer (0.57% and 0.65% for the same samples). It was observed that the cross-linking process reduced the nitrogen content, because the mass of the sample increased due to the addition of GA (0.51% for both cases). Additionally, the results of the XPS analysis of selected chitosan-silica composite material (ChNS_1) and high-resolution XPS spectra of atoms Si2p, O1s and C1s are presented in Appendix A.

The acid–base properties of chitosan–silica materials were determined via potentiometric titration. The dependencies illustrating the change in the density of the surface charge as a function of pH show the diversification of the curves. The surface charge density and the pH of the zero charge point are important parameters characterizing the interface between the surface of the test solid and the electrolyte phase. The charge on the solid surface is created by the reaction between the hydroxyl groups on the surface of the composite and the components of the liquid phase. In the case of aqueous solutions, hydrogen ions (hydroxide) and ions of the carrier electrolyte play an important role in creating the charge at the interface. The point of zero charge (pH_PZC_) determines the number of acid–base sites on the surface of the tested composites, and was determined via joint plotting of the electrolyte titration curve and the experimental curve of suspension. On the other hand, the intersection of the surface charge density curve with the y-axis equals zero. This dependence of the charge density as a function of pH is shown in Appendix A.

The obtained results suggest that the chitosan–silica materials with nanosized silica (ChNS_1 and ChNS_1_GA) show a lower point of zero charges than their silica-gel counterparts (ChSG_1 and ChSG_1_GA), at pH_PZC_ = 5.2 and 5.8 compared with 6.2 and 6.5 values, respectively. In both groups, a slight shift toward higher pH values was observed for the cross-linked samples (a shift of pH_PZC_ ~0.6 and ~0.3 from samples without GA for ChNS_1_GA and ChSG_1_GA samples, respectively). Due to the great attention devoted to adsorption processes in this work, the acid–base behavior is important due to the relationship between the type of generated surface charge and the adsorbate charge in experimental conditions. In the case of electrostatic attraction, the adsorption increase is observed, and its weakening or even lack is found when a repulsive effect prevails. In further analysis of the adsorption process based on the obtained pH_PZC_ values, it was taken into account that under conditions below the indicated values (the pH_PZC_ for ChNS_1, ChNS_1_GA, ChSG_1, and ChSG_1_GA equalling 5.2, 5.8, 6.2, and 6.5), the surface is negatively charged above the pH_PZC_ point, and positively charged below these values.

### 2.2. The Adsorption Properties of the Chitosan–Silica Composites

#### 2.2.1. The Effect of the Chitosan Component on the Adsorption Process

The two main components of synthesized composites are the silica and chitosan. In chemical terms, the biopolymer is characterized by three types of reactive functional groups, i.e., amino, acetamide, and hydroxyl groups. The presence of these groups in a macromolecule enables the generation of intramolecular and intermolecular hydrogen bonds as well as chemical or physical interactions with other substances. Therefore, chitosan is the main component of chitosan–silica composites, and plays a key role in the process of water purification from anionic dyes. Silica, even though it is quantitatively the predominant component of the composite, serves only as a carrier for the organic phase.

Figure 5 shows the adsorption isotherms of the Acid red 88 dye on the composites with different amounts of both components, i.e., with one layer (ChNS_1 and ChSG_1) or a half layer (ChNS_05 and ChSG_05) of chitosan. Comparing the isotherms’ course, it can be seen that the adsorption efficiency correlates with the amount of the biopolymer. The adsorption capacity determined based on the generalized Langmuir equation (GL) for composites with a half and one layer of chitosan synthesized from nanosilica (NS) is 0.28 mmol/g and 0.51 mmol/g, respectively, and from silica gel (SG) is 0.19 mmol/g and 0.31 mmol/g. It can be seen that the use of a double amount of chitosan during the synthesis increases the adsorption capacity for the NS-supported material by 1.8 times, and by 1.6 times for the SG-supported material. This is probably related to the increase in the density of materials being a result of the use of a larger amount of chitosan. This, in turn, means that the structure of samples with one layer of biopolymer is more compact than that of those with a half layer, which reduces the interfacial contact area of adsorbate–adsorbent and diminishes the adsorption capacity towards the pollutant. For comparison, the density of composites with half and one layer of chitosan obtained from NS is 260 g/dm^3^ and 282 g/dm^3^, respectively, and from SG is 290 g/dm^3^ and 301 g/dm^3^. Figure 5 shows that the dye adsorption on the ChNS_05 composite is similar to that on the ChSG_1 composite with double biopolymer content. It follows that the adsorption capacity of composites depends not only on the relative share of organic and inorganic components, but also on their structural and morphological features, which will be discussed in more detail in the next section.

The adsorption process in the discussed systems was carried out under neutral pH conditions; therefore, the mechanism of binding the dye to the composite was based on attractive electrostatic interactions between partially protonated amino groups of chitosan and dye anions, as well as hydrogen bonds involving functional groups of the composite (i.e., amino, amide, hydroxyl, silanol) and dye (azo, sulfonic, hydroxyl functional groups). Depending on the adsorption conditions, especially the ionic strength and pH level, the mechanisms of the process can be changed, including diversification of the attractive/repulsive interactions, the strength of the hydrogen bonding, and partial sharing of complexation.

#### 2.2.2. Effect of Silica Component on the Adsorption Process

Figure 5 presents the isotherms of dye adsorption on the ChNS_1 and ChSG_1 composites. For the synthesis of these materials, the same amount of chitosan (one layer) and different types of silica (nanosilica and silica gel, respectively) were used. The adsorption capacity obtained from optimization with the GL isotherm for the ChNS_1 and ChSG_1 composites was 0.51 and 0.31 mmol/g, respectively. Such significant differences result from the different structural and morphological characteristics of adsorbents, strictly depending on the type of silica component.

In the case of a composite with nanosilica (pyrogenic silica, aerosil), the inorganic matrix takes the form of three-dimensional particles consisting of strongly agglomerated particles of amorphous nanosilica [63]. These particles form a system with a high degree of fractality and a loose structure, which ensures uniform coverage with a thin layer of organic biofilm during mechanosorption in the ball mill. Such structural characteristics of the composite enable free contact of the adsorbate with the biopolymer, and translate into high efficiency of the adsorption process. Moreover, adsorption takes place in the interparticle spaces (voids) of the material. In the case of a composite with silica gel, the matrix is made of porous silica grains. Porous structure and diversification in the diameter and shape of the pores of the solid are an obstacle to the uniform immobilization of the biopolymer. In places wherein a greater amount of chitosan is located, multilayers forms are formed, stabilized by intramolecular interactions, which in turn results in limited access for the adsorbate to the adsorption active centers of the chitosan component (amine, amide, hydroxyl groups). Density is another parameter that is more favorable to dye adsorption on the ChNS_1 composite than on the ChSG_1 composite (282 and 301 g/dm^3^, respectively). A lower density indicates a less compact structure of the solid, which facilitates access to the adsorbate.

In order to comprehensively assess the effectiveness of chitosan–silica composites as adsorbents towards anionic dyes, the equilibrium adsorption studies were supplemented with kinetic measurements. Figure 6 shows changes in the relative adsorbate concentration (c/c_o_) as a function of time (t) and root of time (t^1/2^) for the ChNS_1 and ChSG_1 composites. The comparison of the concentration profiles shows a clear differentiation in the adsorption course: very fast and much slower adsorption for ChNS_1 and ChSG_1, respectively. This may be explained by the differences in their structural and morphological characteristics, and the uniformity of chitosan coverage. In the case of ChNS_1, a large range of the c/c_o_ profiles’ rectilinearity was obtained in the initial stage of the experiment, which proves the high dynamics of the process. This is also confirmed by significant differences in the values of kinetic parameters, which are determined based on a theoretical optimization of the obtained data using the multi-exponential equation (m-exp). For ChNS_1 and ChSG_1, the half-time (t_0.5_) is 1.14 and 33 min, and the rate constant (log k_avg_) is −0.22 and −1.67, respectively. Similar conclusions can be drawn based on the experiment duration necessary to achieve a certain (75%, exemplary) decolorization efficiency of solution (t_75%_). For ChNS_1 and ChSG_1, this parameter is 4 and 640 min, respectively.

#### 2.2.3. The Effect of Cross-Linking on the Adsorption Process

Figure 7 shows the dye adsorption isotherms on the ChNS_1 and ChSG_1 composites, and on the crosslinked materials ChNS_1_GA and ChSG_1_GA, for which the synthesis was extended by the step of cross-linking the chitosan component with glutaraldehyde (GA). The adsorption process was carried out from aqueous solutions with a neutral pH. Comparing the isotherm course, a clear impact of the additional modification on the efficiency of adsorbents in the water purification process is noticeable. The adsorption capacity determined from the GL equation for non-crosslinked and crosslinked composites synthesized with NS is 0.51 and 0.43 mmol/g, respectively, and 0.31 and 0.19 mmol/g with SG. The decrease in the adsorption capacity as a result of modification is related to the mechanism of formation of a three-dimensional polymer network based mainly on covalent imine bonds between chitosan amino groups and GA aldehyde groups. The involvement of some amino groups of the chitosan component in the network’s formation significantly reduces the number of adsorptive active centers available for the adsorbate. The decrease in the performance of the composite containing silica gel is more pronounced compared to the composite containing nanosilica, i.e., by about 40 and 15%, respectively.

The research revealed that the effect of the cross-linking of the chitosan component on the adsorption rate depends on the silica type (Figure 8). In the case of the composite based on NS, the additional modification (using GA) slightly decreased the rate of dye adsorption, which can be explained by the relatively large amount of free adsorptively active groups (after cross-linking) on the outer surface of the composite. Both profiles are characterized by the rectilinearity of c/c_o_ changes in the initial stage of the experiment, which confirms the high dynamics of the adsorption process despite cross-linking. For ChNS_1 and ChNS_1_GA, t_0.5_ = 1.14 and 1.20 min; log k_avg_ = −0.22 and −0.24; t_75%_ = 4 and 6 min and t_90%_ = 35 and 120 min, respectively. For the SG composite, the cross-linking effect on the adsorption kinetics is greater due to the network of internal pores in the solid structure, which can be reduced or blocked by the modifier. For ChSG_1 and ChSG_1_GA, t_0.5_ = 33 and 61 min, and log k_avg_ = −1.67 and −1.94; due to the low adsorption capacity of these composites, the parameter t_75%_ can not be compared, and the same applies to t_90%_.

#### 2.2.4. The Effect of pH on the Adsorption Process

The influence of solution pH on adsorption was studied for the chitosan–silica composite ChNS_1_GA (cross-linked with glutaraldehyde). Analyzing the dye adsorption isotherms in various pH conditions (Figure 9), significant differences in the efficiency of dye removal are found. These differences may be related to the intensity and type of electrostatic interactions between the adsorbate and adsorbent. It is well known that a solution pH determines an adsorbent surface charge and a degree of the adsorbate dissociation/ionization. In the experimental system, the composite surface charge results from all positively and negatively charged surface groups derived from both components, i.e., chitosan and silica. They include (i) hydroxyl, (ii) amino, and (iii) silanol groups. At a lower pH, the composite surface is more positively charged due to the presence of protonated amino groups ($-{NH}_{3}^{+})$. With an increase in the pH, the discussed groups proceed with a gradual deprotonation process into an electrostatically neutral form $\left(-{NH}_{2}\right)$, while the others become ionized, producing a negatively charged composite surface.

Acid red 88 is sodium salt of 4-(2-hydroxy-1-naphthylazo)-1-naphthalene sulfonic acid, so in the pH range of 2–10, its molecules have a negative charge coming from ionized sulfonic acid groups. Due to the complex chemical structure of the dye, its charge also depends on the base–acid properties of the azo and phenolic groups. However, based on the distribution of adsorbate molecular forms as a function of solution pH (Appendix A), it was found that in the experimental pH range, the azo groups’ participation in the total dye charge can be neglected.

Therefore, changes in the adsorbent surface charge and the degree of the adsorbate ionization affected by the solution pH translate into an adsorption efficiency. At a low pH (pH = 2), for which the composite surface is the most positively charged, while adsorbate occurs in form 1 (Appendix A), the highest adsorption was obtained. The mechanism of the process in the studied system is likely ascribable to attractive electrostatic interactions and, to a lesser extent, hydrogen bonds. At pH = 4 and pH = 6, the amounts of adsorbed dye are comparable and significantly lower than those at pH = 2. These differences can be explained by the weakening of the electrostatic attraction between the adsorbate and adsorbent due to partial deprotonation of amino groups. In the pH range of 6–10, there is a successive decrease in the process efficiency as a manifestation of repulsive electrostatic interactions between the negatively charged composite surface (deprotonated hydroxyl and silanol groups) and ionized dye molecules. Here, starting from pH = 6, an intensification of electrostatic forces takes place as a result of a gradual increase in the adsorbate form 4 in the total molecules’ contribution in solution (i.e., the resultant negative charge from both sulfonic and phenolic ionized groups). Ultimately, in the strongly alkaline medium (pH = 10), the dye adsorption is at its lowest level.

The process of adsorption from the electrolyte solution is complicated by the possibility of the formation of the electrical double layer. This phenomenon depends on the ionic strength of the solution, which is equal to the molar concentration of simple electrolytes 1:1 (HCl, NaOH). The effect related to the presence of a double electrical layer on the composite surface and adsorption capacity can be proven through a comparison of the adsorption capacity (from the GL equation) for the ChNS_1_GA at pH = 4 and pH = 6, which is 0.46 and 0.45 mmol/g, respectively. Despite this, at lower pH, when the amount of protonated amino groups is greater, we observe a decrease in the adsorption capacity of the composite due to the additional ionic layer coming from the dissociation of the electrolyte. In a solution with pH = 6, the number of ($-{NH}_{3}^{+})$ groups is lower, but the ionic strength is 100 times weaker than in a solution with pH = 4, which promotes effective adsorption.

Figure 10 shows the concentration profiles for dye adsorption from solutions with different pH levels using the ChNS_1_GA composite. It can be seen that the kinetics depend on both the pH and the ionic strength of the solution. In the initial stage, the process is faster under the following conditions: pH = 2, pH = 4 > pH = 6 > pH = 8 > pH = 10, which is reflected in the values of the rate constant (log k_avg_): 0.20, 0.20, −0.19, −0.27, −0.28, and the half-time (t_0.5_): 0.43, 0.44, 1.08, 1.28, 1.31. It should be emphasized that at this stage, the process is extremely rapid in all systems. Then, the course of the curves begins to diversify, which corresponds well with the differences in the ionic strength of the solutions. The process rate is higher, as follows: pH = 6 > pH = 4 > pH = 2, pH = 8 > pH = 10, for which the times necessary to achieve a process efficiency at the level of 75% and 90% (t_75%_/t_90%_) are as follows: 3.5/41, 7/71, 16/125, 19/125, 228/-- min. The observed changes in adsorption kinetics resulted from (i) slowing down the diffusion of the adsorbate in the bulk solution through electrostatic interactions with ${Cl}^{-}$ or ${OH}^{-}$ electrolyte ions, and (ii) limiting the access of the adsorbate to adsorption active sites through the double electrical layer formed on the solid. In the case of systems at pH = 8 and pH = 10, the effect of the greater ionization of the adsorbate is additionally imposed.

The nanosilica used as a composite component in the raw state consists of spherical nanoparticles with high surface energy. As a result, silica nanoparticles can combine into larger particles, the so-called aggregates. Van der Waals forces, electrostatic forces, and hydrogen bonds are responsible for the adhesion of silica particles to each other. Geometric modification of nanosilica in a ball mill with the addition of solvent results in the rearrangement of particles in aggregates into a more compact form than the original one (i.e., the greater packing of particles) and the formation of higher-order structures, termed agglomerates. Mechanosorption occurs when the mechanical treatment of a solid takes place in the presence a substance showing chemical affinity to a given material. The physical and/or chemical interactions between the organic and inorganic components determine the formation of a new material, a composite. The mechanism of the new material synthesis was based on the electrostatic attraction of chitosan’s amino groups and silica’s silanol groups, and the formation of much weaker hydrogen bonds between chitosan’s amino, acetamino, and hydroxyl groups, and silica’s silanol groups. Due to the specific conditions in the ball mill during the synthesis of chitosan–silica composites (high-energy friction force, elevated temperature, water–gas phase), in addition to physical interactions, chemical interactions (covalent type) in the newly formed structure may be expected. To verify the validity of this thesis, equilibrium adsorption studies were carried out in strongly acidic conditions (pH = 2) on the non-crosslinked ChNS_1 and the crosslinked ChNS_1_GA composites (Figure 11A).

The adsorption capacities of these solids determined from the GL equation are at 0.75 and 0.72 mmol/g. The better ability of the ChNS_1 composite to bind the pollutant proves that the chitosan component is connected to the silica component via strong interactions; otherwise, it would dissolve under the given pH conditions.

To explain the specificity of forming a chemically resistant composite, a short description of the cross-linking method using various cross-linking agents is presented herein. Generally, cross-linking is achieved by the formation of a polymer network based on chemical or physical bonds (covalent and/or ionic) between polymer chains, or between polymer chains and a multifunctional cross-linking agent. Depending on the type of cross-linking agent used, cross-linking takes place with the participation of an amino group or hydroxyl group at the C-6 atom of the polymer chain. In the case of using glutaraldehyde (GA) or genipin (G), the process takes place between chitosan amino groups and the aldehyde groups of GA (covalent imine bond) or the ester groups of G (covalent amine bond), respectively. If epichlorohydrin (ECH) is used, the hydroxyl group of chitosan at the C-6 atom and a reactive oxirane ring of ECH are involved in the process. Another method of cross-linking chitosan is through the addition of negatively charged molecules (complexing inorganic ions, sulfuric acid, etc.) or macromolecules (alginate, pectin, hyaluronic acid, etc.). The type of cross-linker and its concentration, the average molecular weight of chitosan and its deacetylation degree, the concentration of chitosan solution, and the cross-linking time affect the final product density. Chitosan can also undergo cross-linking under the influence of electron beam, gamma radiation, UV radiation, enzymes, or heating [64,65,66,67]. The specific conditions in the ball mill during the synthesis of the chitosan–silica composite (ChNS) suggest that the thermally induced crosslinking of chitosan took place, providing chemical stability to the solid during equilibrium adsorption in a strongly acidic solution. The strong aggregation and agglomeration of nanosilica particles containing $-{OH}^{-}$ groups can be treated here as a polyanionic substance, and chitosan may be treated as a polycationic one. On the other hand, the formation of covalent bonds (Si-O-C) between polymer carbinol groups and silica silanol groups can be expected. Thus, the considered cross-linking process is based on a thermally induced mechanism of ionic and/or covalent interactions. Additonally, structure characterization of chitosan and exemplary chitosan-silica nanocomposite (ChNS_1) by FTIR spectra (Appendix A) is described in Appendix A.

In Figure 11B, the kinetic profiles for AR88 adsorption on the pure chitosan (Ch), nanosilica (NS), and nanosilica-based composite (ChNS_1) are presented. The applied weights of Ch and NS correspond to their amounts in ChNS_1, which were calculated based on the synthesis procedure. A much faster adsorption on the composite, compared to that on the chitosan, was observed, while adsorption on the nanosilica did not occur. This proves the synergistic effect of the presence of the dye-binding groups of the chitosan component, and the developed interfacial surface of nanosilica component in the composite. 

#### 2.2.5. The Effect of Temperature on the Adsorption Process

Temperature is a parameter that can affect significantly adsorption capacity. The direction of changes in the adsorption properties depends on the thermodynamics of the process. When the temperature increases, the adsorption capacity of the material toward the pollutant also increases; the process is regarded as endothermic, and when the opposite trend is observed, the process is exothermic.

In Figure 12, the adsorption isotherms of Acid red 88 from aqueous solutions on the composites ChNS_1 and ChNS_1_GA in the temperature range of 5–45 °C are shown. Comparing the experimental data, a strong decrease in adsorption with an increase in temperature is observed. At extreme temperatures (5 and 45 °C), the adsorption capacities for ChNS_1 differ by about 14%, while for ChNS_1_GA, they differ by about 17%. The observed direction of changes in adsorption values indicates the exothermic nature of the process, which can be explained by (i) an increase in the adsorbate solubility due to greater interactions with solvent molecules, accompanied by the formation of water clusters; and (ii) an increase in the oscillation energy of adsorbate molecules and the resultant partial desorption of the adsorbate from the surface of the solid to the bulk phase.

The influence of temperature on the adsorption process in the studied systems was also presented as a function of lnK_c_ on (1/T) (Figure 13). Based on parameters of this dependence and the suitable equations, the values of thermodynamic functions, i.e., enthalpy, entropy, and Gibbs free energy, were estimated (Table 3). The negative values of enthalpy and Gibbs free energy suggest that the adsorption process was exothermic in nature and spontaneous. The order of enthalpy magnitude indicates that the adsorbate molecules are connected to the composite surface by physisorption.

The effect of temperature on the adsorption rate was studied for the system ChNS_1_GA–AR 88. An analysis of the kinetic profiles indicates that adsorption is a fast and temperature-dependent process (Figure 14). An increase in the temperature of the adsorption system accelerates the water purification process, which can be explained by the increase in the kinetic energy of the adsorbate molecules and their faster diffusion to the interfacial surface. It was found that regardless of the process temperature, in the initial period of the experiment, the course of adsorption is so rapid that after only 0.77–2.17 min (half time, t_0.5_), 50% of the dye was removed. Then, the process slows down a bit, and differences between the curves obtained at different temperatures become noticeable. At the same time, increasing the temperature of the system from 35 to 45 °C has almost no effect on the rate of the adsorption process. The rate constant (log k_avg_) for the dye adsorption process at 45, 35, 25, and 15 °C is −0.05, −0.08, −0.24, and −0.50, respectively. For the systems at extreme temperatures, i.e., 45 and 15 °C, the time after which a process efficiency of 75% was obtained was 2 and 41 min, respectively (representing a 7-fold difference); when an efficiency of 90% was obtained, the times were 14 and 173 min (representing a 12-fold difference).

#### 2.2.6. Adsorption Data Optimization

The experimental data of AR 88 adsorption on the chitosan–silica composites were optimized using the generalized Langmuir (GL) equation [68,69,70,71]:(1)$$\theta ={\left[\frac{{\left(Kc_{eq}\right)}^{n}}{1+{\left(Kc_{eq}\right)}^{n}}\right]}^{\frac{m}{n}}$$
where θ = a_eq_/a_m_ is the global adsorption isotherm; a_m_ is the adsorption capacity; m, n arethe heterogeneity parameters; and K is the equilibrium constant.

Depending on the values of the m and n parameters, the GL equation is reduced to the following simpler forms: the Langmuir isotherm (L) (m = n = 1); the Langmuir–Freundlich isotherm (LF) (0 < m = n ≤ 1); the generalized Freundlich isotherm (GF) (n = 1, 0 < m ≤ 1); and the Tóth (T) isotherm (m = 1, 0 < n ≤ 1).

In analyzing the values of the optimization parameters (Table 4), one can see diversity in the energetic heterogeneity of most systems (m~0.14–1; n~0.37–1). The high values of equilibrium constant (log K~2–2.67) mean relatively strong adsorption. All adsorption isotherms display good agreement between the experimental points and fitted lines, which is confirmed by the SD(a) and R^2^ values.

Kinetic data of AR 88 adsorption on composites were optimized using the multi-exponential equation, which is a generalization of the Lagergren equation to a series of parallel first-order processes [69,71,72,73,74,75]:(2)$$c=\left(c_{o}-c_{eq}\right)\sum_{i=1}^{n}f_{i}exp\left(-k_{i}t\right)+c_{eq}$$
where i is the term of the m-exp equation, k_i_ is the rate coefficient, and f_i_ is the contribution of the rate coefficient.

In Table 5, the following selected parameters of applied equation are presented: the logarithm of the average rate constant (log k_avg_), the average adsorption half-time (t_0.5_), the time for 75% or 90% decoloration efficiency (t_75%_/t_90%_), and the relative adsorbate uptake (u_eq_). The average adsorption half-time (t_0.5_) is calculated numerically. In Appendix A, the following full parameters for exemplary system (AR 88/ChNS_1_GA) are collected: the contribution of the rate coefficient in the complete process (f_i_), the logarithm of the rate coefficient (log k_i_), and the adsorption half-time for the respective equation term (t_0.5,i_). The latter parameter is determined from the respective rate coefficient t_0.5,i_ = (ln 2)/k_i_. Graphically, the distribution of the rate coefficients of the respective terms in the m-exponential equation is shown in Appendix A.

For all the studied systems, three terms in the exponential equation are the optimal number to give a high quality of kinetic fitting, as confirmed by the values of the determination coefficient 1 − R^2^ (2.00 × 10^−4^ − 3.04 × 10^−3^) and the relative standard deviation SD(c)/c_o_ (0.22–0.65%). I propose: Moreover, in Figure 6, Figure 8, Figure 10 and Figure 14, there are no systematic deviations of relative concentration (Δc_i_/c_o_) for the experimental data and the fitting lines, that confirms the correctness of selection a theoretical model for description of the adsorption kinetics.

## 3. Materials and Methods

### 3.1. Materials and Chemicals

Chitosan from shrimp shells with a quality level QL = 200, a molecular weight from 190,000 to 370,000 Da, and a degree of deacetylation of around 75% was purchased from Sigma-Aldrich (Darmstadt, Germany). Fumed nanosilica A-300 with a specific surface area of 320 m^2^/g was obtained from the Chuiko Institute of Surface Chemistry (Kalush, Ukraine). Silica gel C-35 with a specific surface area of ~550 m^2^/g, a molar mass (M) 60.09 g/mol, and a density of 2.2 g/cm^3^ and glutaraldehyde solution (25%) were purchased from Sigma-Aldrich (Darmstadt, Germany). Ultrapure water produced by Millipore Simplicity (Merck-Millipore, Molsheim, France) with a UV device was used for the synthesis of composites. Acid red 88 dye with a purity of 75% was purchased from Sigma-Aldrich (Darmstadt, Germany); hydrochloric acid concentrate 0.1 M and sodium hydroxide concentrate 0.1 M purchased from Chempur (Gliwice, Poland) were used for the adsorption experiments.

### 3.2. Synthesis of Chitosan–Silica Composites

The preparation of chitosan–nanosilica composites (ChNS) and chitosan–silica gel composites (ChSG) consisted of the geometrical modification of silica and the mechanosorption of chitosan. The process was carried out as follows: 25 g of silica (nanosilica A-300 or silica gel C-35/05 Y2) and a calculated amount of chitosan as a solution in 18 g of water were placed into a ball mill (volume 0.5 dm^3^) equipped with ceramic balls with a diameter of 2 cm. Mechanical treatment at a rotation speed of 60 rev/min was carried out for 2 h. Depending on the amounts of the components used (Table 6), composites with one layer (ChNS_1 and ChSG_1) or a half layer (ChNS_05 and ChSG_05) of chitosan were obtained. Some of the samples of ChNS_1 and ChSG_1 were subjected to the crosslinking process using glutaraldehyde (GA). For this purpose, 2 g of a 25% wt. aqueous solution of aldehyde was placed into the ball mill, and then mechanical treatment was continued for 1 h. The cross-linked samples were marked as ChNS_1_GA and ChSG_1_GA. All the obtained powdered composites were dried at 40 °C for 5 h. The list of materials along with the corresponding bulk density is summarized in Table 6.

### 3.3. Methods of Characterization of Chitosan–Silica Composites

#### 3.3.1. Small-Angle X-ray Scattering (SAXS)

The SAXS analysis was carried out via X-ray diffraction (XRD) using an Empyrean diffractometer (PANalytical, Malvern, UK) and Cu anode X-ray tube radiation using the SAXS/WAXS sample stage with capillary mode. The device was powered by a 4 kW high-voltage X-ray generator and generator settings of 40 kV and 40 mA. The incident beam path consisted of W/Si, and a graded X-ray mirror with an elliptic shape. The SAXS configuration includes a 2θ range of −0.1–4 degrees of 2θ, with a 0.005 step size and a counting time of 1.76 (s) and 821 points for every single scan. The primary beam was measured using a beam attenuator of Cu 0.2 mm. The measurements were taken using a PIXcel3D detector and a receiving slit with a 0.05 mm active length. The length of the scattering vector (or scattering vector) q is defined as q = 4πsinθ/λ, where 2θ is the scattering angle, and λ is the X-ray wavelength (1.5418 Å). Background scattering was performed by an air-scattering measure with an empty sample holder. Dv(R) calculations were performed using the indirect Fourier transformation technique applied in EasySAXS software, version 2.0a. In this case, the applied algorithm is based on Tikhonov’s regularization method. The pair distance distribution function (PDDF), as real space counterparts of the experimental intensity (p(r)), was calculated using the EasySAXS ((PANalytical), UK) program, as an indirect Fourier transform of these data. The Guinier plot, ln(I(q)) vs. q_2_, was used to determine the radius of gyration Rg from the slope of the plot (−Rg^2/3^).

#### 3.3.2. Other Techniques

The surface charge properties were determined via potentiometric titration of the suspension with a 765 Dosimat Autoburette (Metrohm, Herisau, Switzerland) combined with a PHM240 pH-meter (Radiometer, Copenhagen, Denmark). A constant temperature of 25 °C (provided by the thermostat Ecoline RE207, Lauda, Germany) and computer program (Titr_v3, written by Marczewski and Janusz, Faculty of Chemistry, Maria Curie-Sklodowska University, Lublin, Poland) were applied during experiments. The other experimental details are as follows: 30 mL of 0.1 mol/L NaCl as an electrolyte, 0.3 mL of 0.5 mol/L HCl as an initial pH stabilizer, 0.05 g of each solid, and 0.2 mol/L NaOH as a titrant. The measurements were supported by protecting the gas atmosphere (nitrogen) to defeat the problem of carbon dioxide contamination.

The textural properties of all the chitosan–silica composites were estimated from low-temperature nitrogen adsorption–desorption isotherms measured at 77 K over the whole range of relative pressures (from 0 to 950 mmHg), using a sorption analyzer (ASAP 2020, Micromeritics, Norcross, GA, USA). The specific surface area (S_BET_) was calculated from experimental isotherms according to the standard BET method. The pore size distribution curves were obtained from the desorption branch of the isotherm using the Barrett–Joyner–Halenda (BJH) model with cylindrical pores and Faas correction. The total pore volume (V_t_) was obtained from the amount of nitrogen adsorbed at P/P𝑜 = 0.99. The pore volume of micropores (V_mic_) was estimated using the t-plot method. Before the analysis, all samples were outgassed at 90 °C and a pressure of 1 mmHg for 24 h in the degas port of the analyzer.

The morphology of the obtained materials was investigated using a scanning electron microscope (SEM), QuantaTM 3D FEG (FEI Company, Hillsboro, OR, USA), and the morphological characteristics are presented in the SEM images.

A carbon, hydrogen, and nitrogen analysis of the chitosan–silica composites was carried out using a Series II CHNS/O Analyzer 2400 (Perkin Elmer, Waltham, MA, USA). The temperature of the reduction and the combustion processes was 650 and 950 °C, respectively. A 500 mg sample of each composite was used during analysis.

#### 3.3.3. Adsorption Equilibrium

Commercial Acid red 88 (AR 88) was taken as a model anionic dye for adsorption experiments. The compound is characterized by the following physicochemical properties: a molecular weight of 400.38 g/mol, an ionization constant of 11.06, and a water solubility of 0.4%. AR 88 adsorption studies on the chitosan–silica composites were performed using the static method with UV–Vis spectrophotometric measurements (Cary 4000, Varian Inc., Mulgrave, Australia). The studied adsorbents (0.05 g) were contacted with AR 88 solutions at various concentrations (0.06–2.6 mmol/L) in neutral (ultrapure water as a solvent) or at fixed pH conditions (2, 4, 6, 8, 10). The Erlenmeyer flasks with adsorption systems were placed in a shaker (New Brunswick Scientific Innova 40R Model) and agitated at a fixed temperature for 48 h (110 rpm; 5, 15, 25, 35, and 45 °C). When equilibrium was reached, the equilibrium concentrations and the adsorbed amounts of adsorbate were determined from the maximum absorbance of spectra (λ = 503 nm) and the mass balance equation, respectively.

#### 3.3.4. Adsorption Kinetics

Adsorption kinetic measurements were performed using a UV-Vis spectrophotometer (Cary 100, Varian Inc., Australia) equipped with a flow cell. Dye solution with an initial concentration of 0.095 mmol/L (neutral or fixed pH, i.e., pH = 2, 4, 6, 8, 10) and a volume of 200 mL was contacted with 0.1 g of chitosan–silica composite in a thermostatic vessel (Ecoline RE 207, Lauda, Germany). During the adsorption process, the suspension solution was stirred with a mechanical stirrer (110 rpm) and thermostated at fixed temperatures (15, 25, 35, 45 °C). At a certain time, solution samples were collected in a flow cell after passing through a glass wool filter, and absorption spectra were measured. After each measurement, the solution sample was returned to the reaction vessel. The obtained absorption spectra were used for the determination of the profiles of relative concentration vs. time, and the profiles of relative concentration vs. square root of time for the studied systems. Applying an automated procedure for sampling and prompt spectra measuring resulted in kinetic profiles with a large number of experimental points (47–87).

## 4. Conclusions

Chitosan materials are interesting due to their unique properties in terms of physicochemistry, renewability, and biocompatibility. The paper presents a series of chitosan–silica composites differentiated by their silica content, the size of the biopolymer layer, and their cross-linking with glutaraldehyde. A novel synthetic route using mechanochemical synthesis was proposed. Due to the specific conditions in the ball mill during the synthesis of the chitosan–silica composites (high-energy friction force, elevated temperature, water–gas phase), chemical bonds (of the covalent type) are probably formed.

The adsorption capacity of the composites depends on their relative share of organic and inorganic components, and their structural and morphological features. The better adsorption capacity of the ChNS_1 composite to bind anionic acid Red 88 dye at a low pH proves that the chitosan component is connected to the silica component by strong interactions without the effect of dissolving chitosan (which occurs in such low conditions for individual biopolymers).

The influence of an important parameter (i.e., pH) in conditioning the adsorption process was also determined. The conditions of low pH (pH = 2), in which the composite surface is the most positively charged, ensure the highest adsorption efficiency of the ionic form of the dye. Adsorption efficiency correlates with the amount of biopolymer. The adsorption capacity for composites with half and one layer of chitosan (NS) is 0.28 and 0.51 mmol/g, respectively, and for composites with half and one layer of SG is 0.19 and 0.31 mmol/g. The adsorption kinetics of AR88 dye is very fast for ChNS_1, and much slower for ChSG_1 (silica-type effect). For ChNS_1 and ChSG_1, the half-time (t_0.5_) was determined to be 1.14 and 33 min, with the rate constant (log k_avg_) equalling −0.22 and −1.67, respectively. The high degree of fractality and the loose NS structure ensure uniform coverage with a thin layer of biopolymer, and allow free contact of the adsorbate with the active sites, which promotes the high efficiency of the adsorption process. Adsorption of the AR88 dye on the proposed composites is also dependent on temperature. An increase in the temperature of the adsorption system accelerates the adsorption process, which is associated with an increase in the kinetic energy of the adsorbate molecules, and with their faster diffusion to the surface active sites. Based on the temperature dependences, log k_avg_ for the dye adsorption on the cross-linked sample at 45, 35, 25, and 15 °C equals −0.05, −0.08, −0.24, −0.50, respectively. For the temperatures of 45 °C and 15 °C, the time after which a process efficiency of 75% was obtained was 2 and 41 min (representing a 7-fold difference), and the time taken for an efficiency of 90% was 14 and 173 min (representing a 12-fold difference). Research will be continued in order to obtain other effective environmentally friendly composites based on natural biopolymers.

## Figures and Tables

**Figure 1 ijms-24-11818-f001:**
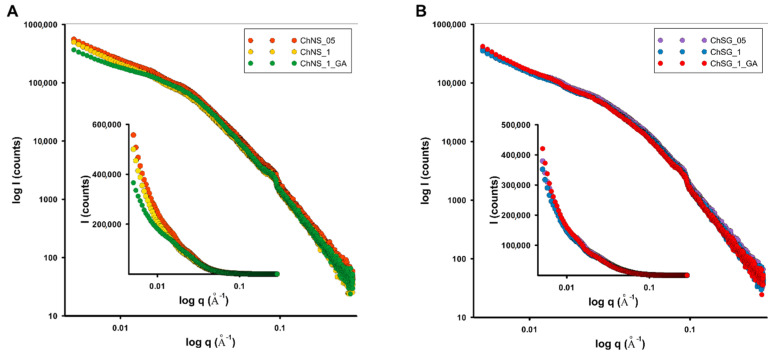
The SAXS patterns for the investigated composites: composites based on nanosilica (NS): ChNS_05, ChNS_1, ChNS_1_GA (**A**), and composites based on silica gel (SG): ChSG_05, ChSG_1, ChSG_1_GA (**B**).

**Figure 2 ijms-24-11818-f002:**
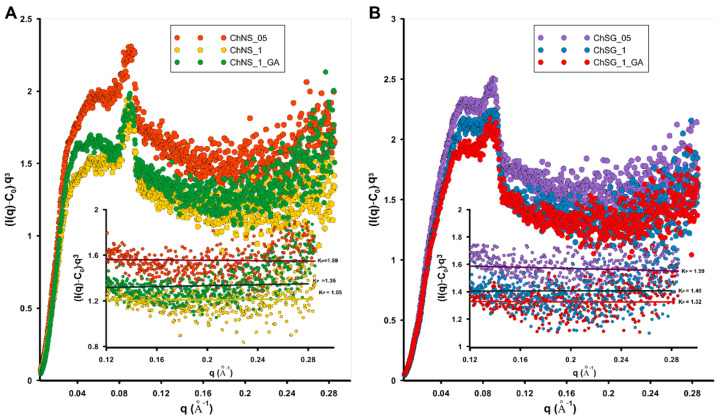
Porod plots with slit collimation calculated for the investigated silica–biopolymer composites ChNS_05, ChNS_1 and ChNS_1_GA (**A**) and ChSG_05, ChSG_1 and ChSG_1_GA (**B**). The asymptotic range of each curve was approximated using a linear function, and a specific Porod constant was indicated for each function.

**Figure 3 ijms-24-11818-f003:**
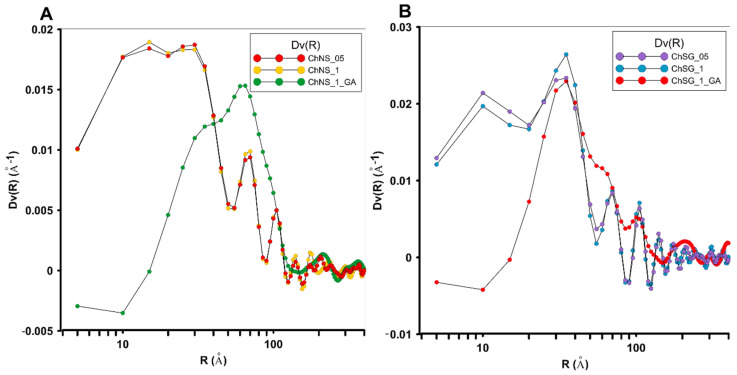
Comparison of volume size distribution function of the scattering heterogeneities in composites based on nanosilica: ChNS_05, ChNS_1, ChNS_1_GA (**A**), and based on silica gel: ChSG_05, ChSG_1, ChSG_1_GA (**B**).

**Figure 4 ijms-24-11818-f004:**
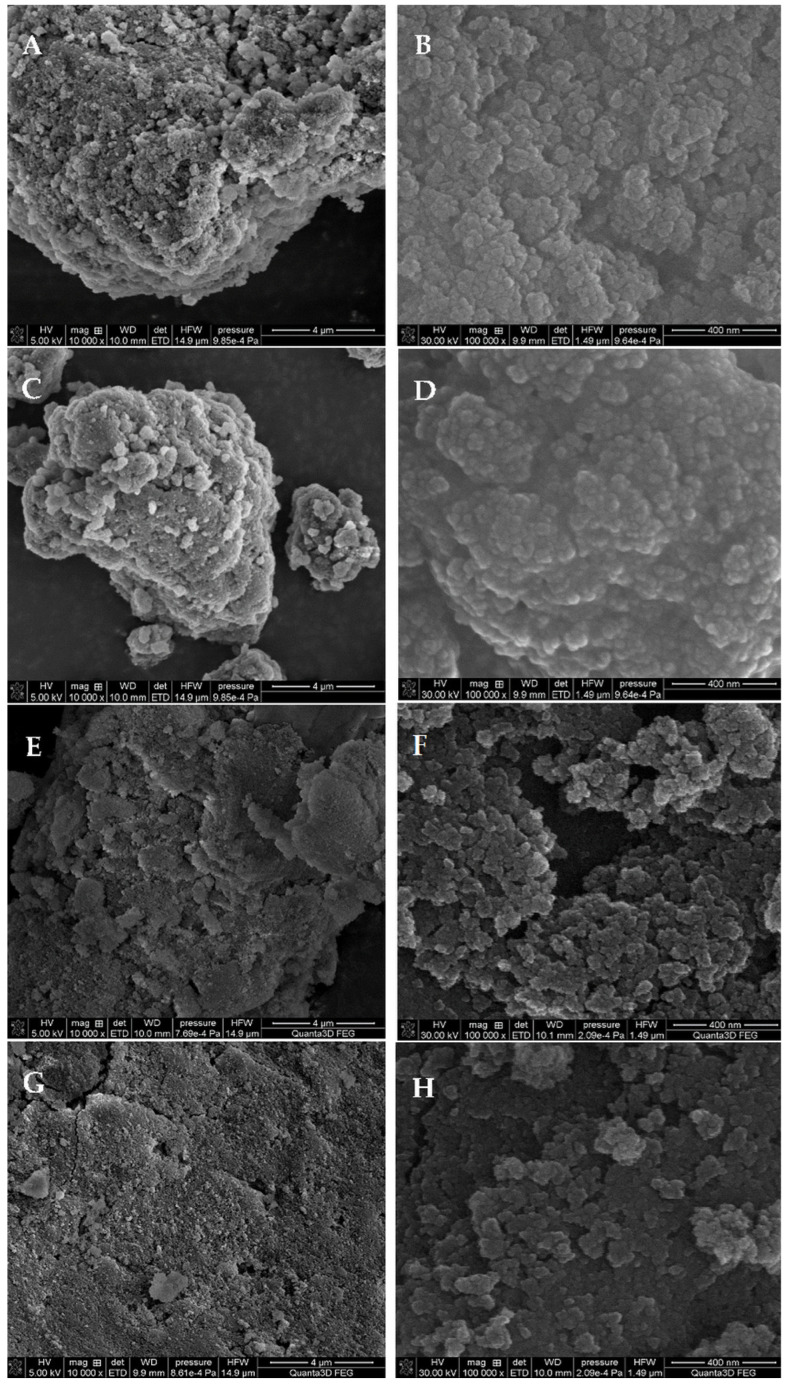
SEM images of the selected chitosan–silica composites: ChNS_1 (**A**,**B**), ChNS_1_GA (**C**,**D**) and ChSG_1 (**E**,**F**) and ChSG_1_GA (**G**,**H**), at different magnifications.

**Figure 5 ijms-24-11818-f005:**
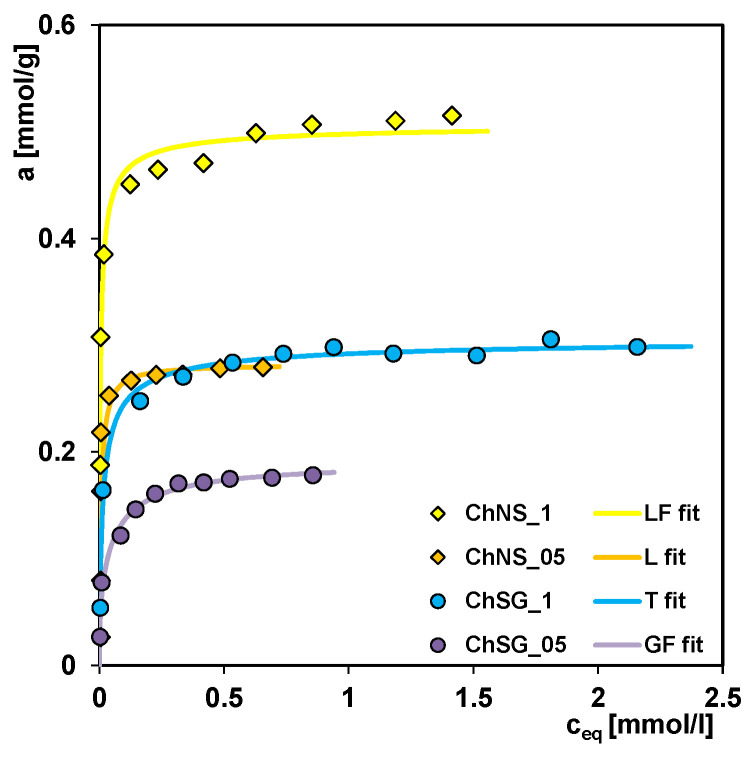
Adsorption isotherms of AR 88 on the composites with different compositions: ChNS_1, ChSG_1, ChNS_05, and ChSG_05.

**Figure 6 ijms-24-11818-f006:**
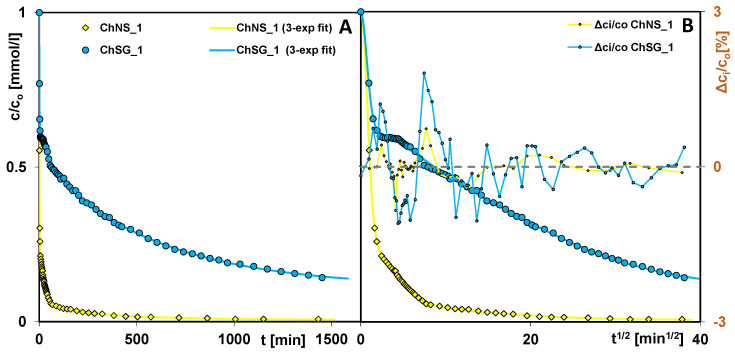
Comparison of AR 88 adsorption kinetics on the composites ChNS_1 and ChSG_1 as a function of time (**A**), and root of time (**B**). The lines correspond to the fitted multi-exponential equation.

**Figure 7 ijms-24-11818-f007:**
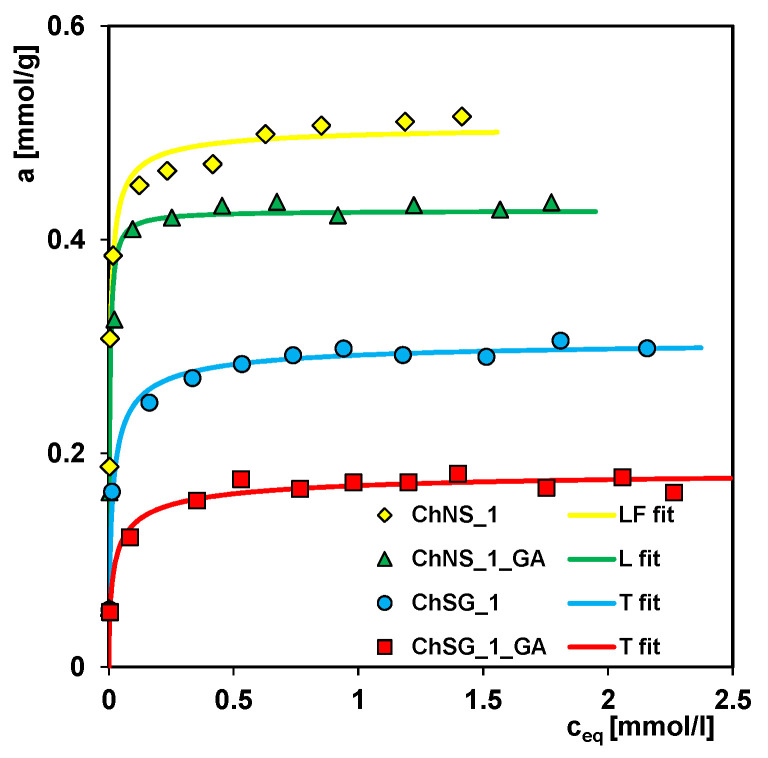
The isotherms of dye adsorption on the ChNS_1, ChSG_1, ChNS_1_GA, and ChSG_1_GA composites. The lines correspond to the fitted multi-exponential equation.

**Figure 8 ijms-24-11818-f008:**
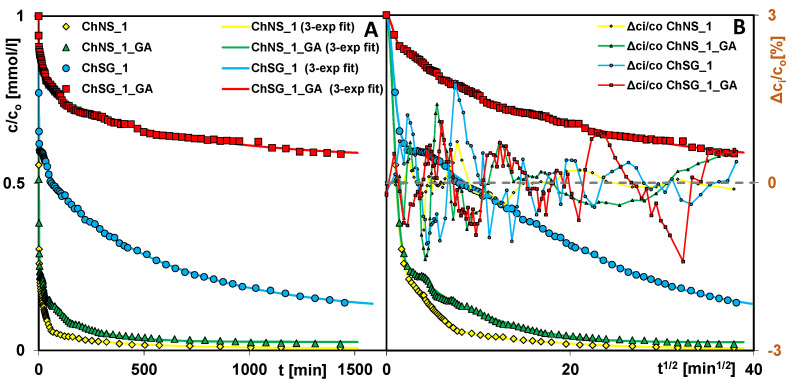
Comparison of AR 88 adsorption kinetics on the ChNS_1, ChSG_1, ChNS_1_GA, and ChSG_1_GA composites as a function of time (**A**), and root of time (**B**). The lines correspond to the fitted multi-exponential equation.

**Figure 9 ijms-24-11818-f009:**
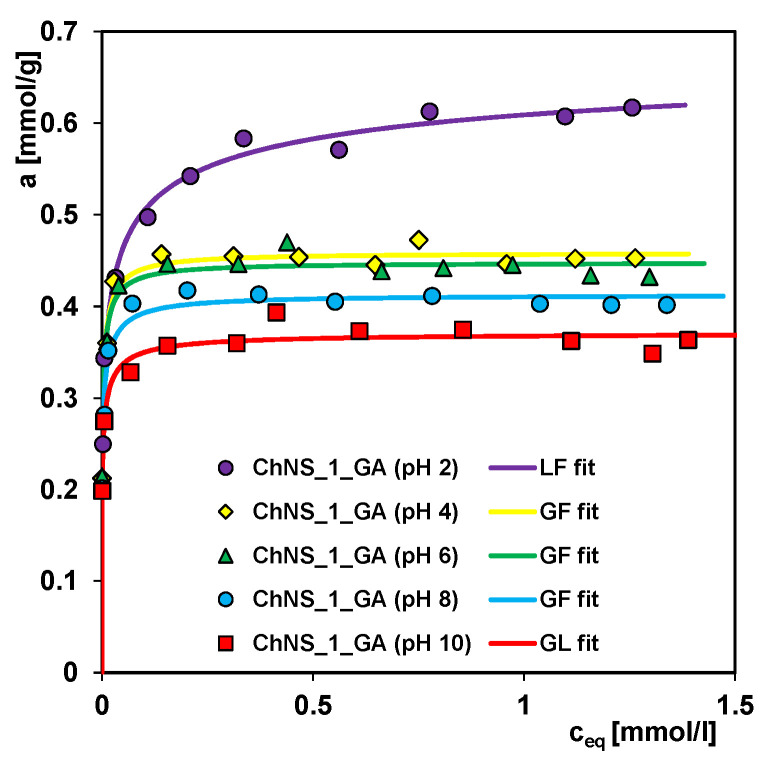
Adsorption isotherms of Acid red 88 on the chitosan–silica composite ChNS_1_GA (cross-linked with glutaraldehyde) under different pH conditions.

**Figure 10 ijms-24-11818-f010:**
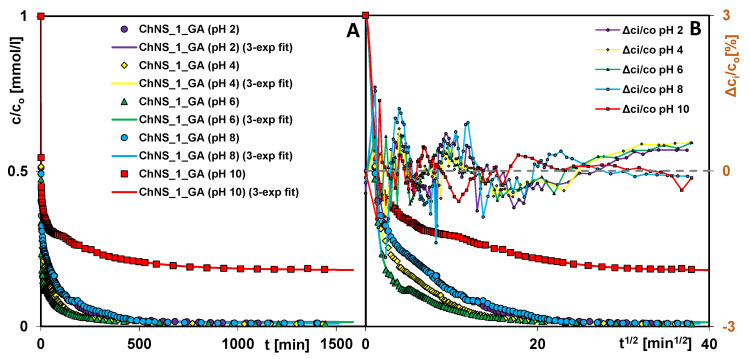
Comparison of AR 88 adsorption kinetics on the composite ChNS_1_GA at various pH levels as a function of time (**A**) and root of time (**B**). The lines correspond to the fitted multi-exponential equation.

**Figure 11 ijms-24-11818-f011:**
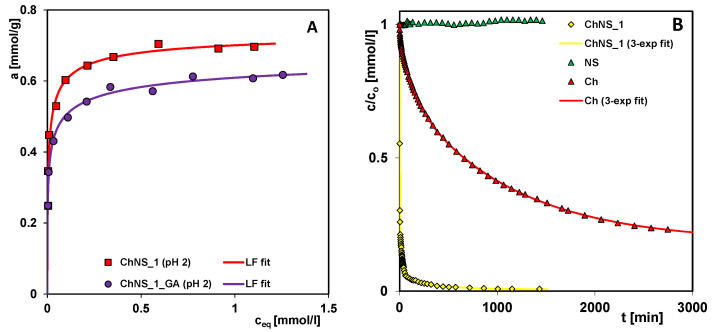
Adsorption isotherms under strongly acidic conditions (pH = 2) on the non-crosslinked ChNS_1 and the crosslinked ChNS_1_GA composites (**A**); Kinetic profiles for AR88 adsorption on the pure chitosan (Ch), nanosilica (NS), and nanosilica-based composite (ChNS_1) (**B**).

**Figure 12 ijms-24-11818-f012:**
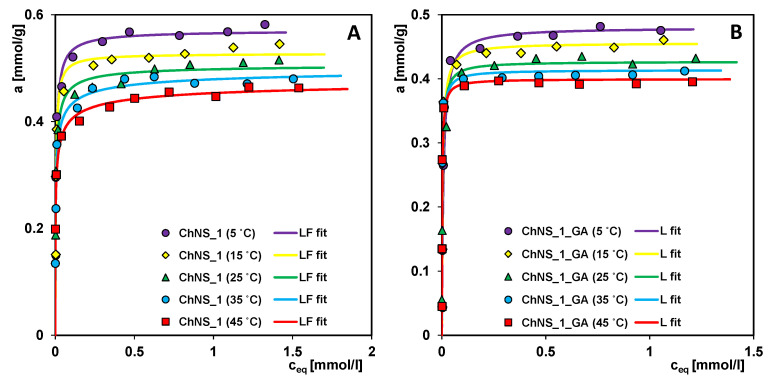
Adsorption isotherms of Acid red 88 from aqueous solutions on the composite ChNS_1 (**A**) and ChNS_1_GA (**B**) in the temperature range of 5–45 °C.

**Figure 13 ijms-24-11818-f013:**
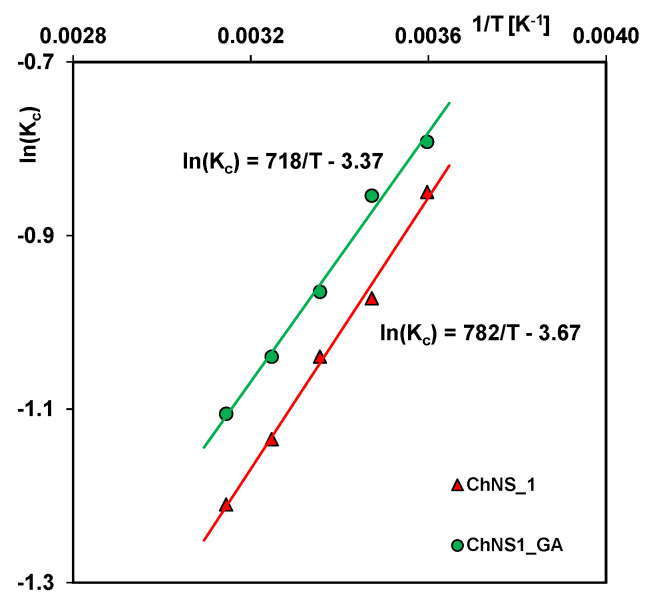
Van’t Hoff plot for Acid red 88 adsorption from aqueous solutions on the composite ChNS_1 and ChNS_1_GA.

**Figure 14 ijms-24-11818-f014:**
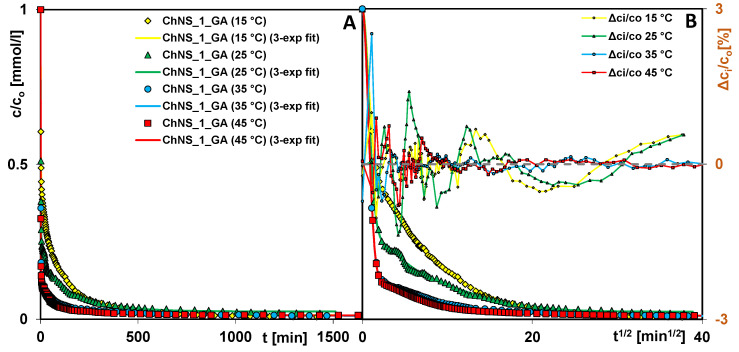
Comparison of AR 88 adsorption kinetics on the composite ChNS_1_GA at various temperatures as a function of time (**A**) and root of time (**B**). The lines correspond to the fitted multi-exponential equation.

**Table 1 ijms-24-11818-t001:** Comparison of the structural parameters of Porod approximation and the S_SAXS_ interfacial area as the porous features of the investigated systems, with the parameters obtained from low-temperature nitrogen sorption data. The surface properties, as pH_PZC_, are summarized in the last column.

Sample	Porod Approximation				S_SAXS_ ^e^ [m^2^/g]	Surface Area (S_BET_) [m^2^/g]		Pore Volume [cm^3^/g]		pH_PZC_
	K_P_ ^a^	Q ^b^ [Å^−1^]	C_0_ ^c^	S/V ^d^ [Å^−1^]		S_BET_Total_ ^f^	S_MIC_ ^g^	V_Total_ ^h^	V_MIC_ ^i^	
ChNS_05	1.59	88	−28	0.072	277	244	3.6	0.93	0.001	-
ChNS_1	1.05	67	−23	0.062	219	209	26.0	0.84	0.013	5.2
ChNS_1_GA	1.35	69	−35	0.078	227	214	9.5	0.71	0.003	5.8
ChSG_05	1.59	81	−27	0.078	269	257	-	0.83	-	-
ChSG_1	1.40	77	−27	0.072	239	201	8.0	0.70	0.003	6.2
ChSG_1_GA	1.32	78	−23	0.067	190	188	27.7	0.65	0.01	6.5

^a^ Porod constant, ^b^ Scattering invariant *Q* is proportional to the mean-square density fluctuation of scattering volume. *Q* = 2π^2^·Δ*ρ*^2^·*V* where volume *V* and scattering contrast Δ*ρ,*
^c^ Bacground constant which illustrates asymptotic decay of the SAXS curve at the high q values, ^d^ The surface-to-volume ratio from the ratio of the Porod constant K_P_ to the Porod invariant Q, S/V = 4K_P_/Q; ^e^ Surface area by SAXS calculated by $S_{SAXS}=\frac{10000\cdot \frac{S}{V\;}[{Å}^{-1}]}{d\;[\frac{g}{{cm}^{3}}]},$ where *d* is the bulk density of the material; ^f^ The S_BET_ BET surface area was calculated using experimental points at a relative pressure of (P/P_0_) 0.035–0.31, where P and P_0_ are denoted as the equilibrium and saturation pressure of nitrogen; ^g^ S_MIC_, the micropore surface area calculated using the t-plot method with fitted statistical thickness in the range of 3.56 to 4.86 Å, ^h^ Vt is the total pore volume calculated using 0.0015468 of the amount of nitrogen adsorbed at P/P_0_ = 0.99, ^i^ V_MIC_ the micropore volume by t-plot.

**Table 2 ijms-24-11818-t002:** Percentage of elements for the tested composites.

Adorbent	% C	% H	% N	% N/C	% N/H	% N/C + H
ChNS_05	2.04	0.51	0.28	0.14	0.55	0.11
ChNS_1	3.14	0.68	0.57	0.18	0.84	0.15
ChNS_1_GA	4.33	0.48	0.51	0.12	1.06	0.11
ChSG_05	2.31	0.58	0.29	0.13	0.50	0.10
ChSG_1	4.36	0.82	0.65	0.15	0.79	0.13
ChSG_1_GA	4.63	0.77	0.51	0.11	0.66	0.09

**Table 3 ijms-24-11818-t003:** Thermodynamic functions, i.e., enthalpy, entropy, and Gibbs free energy for the ChNS_1 and ChNS1_GA composites.

Adsorbent (T)	ΔG [kJ/mol]	ΔS [kJ/mol × K]	ΔH [kJ]	R^2^
ChNS_1 (278K)	−14.00	0.0270	−6.50	0.995
ChNS_1 (288K)	−14.21	0.0268		
ChNS_1 (298K)	−14.54	0.0270		
ChNS_1 (308K)	−14.78	0.0269		
ChNS_1 (318K)	−15.06	0.0269		
ChNS_1_GA (278K)	−14.14	0.0294	−5.97	0.992
ChNS_1_GA (288K)	−14.50	0.0296		
ChNS_1_GA (298K)	−14.72	0.0294		
ChNS_1_GA (308K)	−15.03	0.0294		
ChNS_1_GA (318K)	−15.34	0.0295		

**Table 4 ijms-24-11818-t004:** The parameters of the generalized Langmuir equation for the investigated systems.

Adsorbent (Fit)	a_m_ [mmol/g]	m	n	log K	R^2^	SD (a)
ChNS_05 (L)	0.28	1	1	2.19	0.73	0.051
ChNS_1 (LF)	0.51	0.69	0.69	2.45	0.96	0.024
ChNS_1_GA (L)	0.43	1	1	2.46	0.98	0.020
ChSG_1_05 (GF)	0.19	0.39	1	1.85	0.98	0.008
ChSG_1 (T)	0.31	1	0.64	2.23	0.99	0.010
ChSG_1_GA (T)	0.19	1	0.54	2.26	0.96	0.008
ChNS_1, pH 2 (LF)	0.75	0.51	0.51	2.17	0.96	0.038
ChNS_1_GA, pH 2 (LF)	0.72	0.37	0.37	1.99	0.99	0.015
ChNS_1_GA, pH 4 (GF)	0.46	0.18	1	1.81	0.98	0.012
ChNS_1_GA, pH 6 (GF)	0.45	0.17	1	1.73	0.97	0.013
ChNS_1_GA, pH 8 (GF)	0.41	0.14	1	1.38	0.96	0.015
ChNS_1_GA, pH 10 (GL)	0.37	0.18	0.81	1.64	0.94	0.016
ChNS_1, 5 °C (LF)	0.57	0.84	0.84	2.34	0.92	0.046
ChNS_1, 15 °C (LF)	0.53	0.94	0.94	2.43	0.92	0.041
ChNS_1, 25 °C (LF)	0.51	0.69	0.69	2.45	0.96	0.024
ChNS_1, 35 °C (LF)	0.50	0.49	0.49	2.56	0.98	0.019
ChNS_1, 45 °C (LF)	0.49	0.45	0.45	2.52	0.99	0.010
ChNS_1_GA, 5 °C (L)	0.48	1	1	2.11	0.90	0.053
ChNS_1_GA, 15 °C (L)	0.46	1	1	2.34	0.88	0.054
ChNS_1_GA, 25 °C (L)	0.43	1	1	2.46	0.98	0.020
ChNS_1_GA, 35 °C (L)	0.41	1	1	2.55	0.87	0.050
ChNS_1_GA, 45 °C (L)	0.40	1	1	2.67	0.87	0.049

**Table 5 ijms-24-11818-t005:** Comparison of the selected parameters of the multi-exponential equation.

Adsorbent	log k_avg_	t_0.5_ [min]	t_75%_/t_90%_ [min]	u_eq_	SD(c/c_o_) [%]	1 − R^2^
ChNS_1	−0.22	1.14	4/35	0.99	0.25	2.00 × 10^−4^
ChNS_1_GA	−0.24	1.20	6/120	0.98	0.59	1.16 × 10^−3^
ChSG_1	−1.67	33	640/--	0.89	0.65	1.20 × 10^−3^
ChSG_1_GA	−1.94	61	--/--	0.43	0.52	3.04 × 10^−3^
ChNS_1_GA, pH 2	0.20	0.43	16/125	0.98	0.45	7.58 × 10^−4^
ChNS_1_GA, pH 4	0.20	0.44	7/71	0.99	0.31	4.92 × 10^−4^
ChNS_1_GA, pH 6	−0.19	1.08	3.5/41	0.99	0.37	7.69 × 10^−4^
ChNS_1_GA, pH 8	−0.27	1.28	19/125	0.99	0.61	1.80 × 10^−3^
ChNS_1_GA, pH 10	−0.28	1.31	228/--	0.82	0.31	7.20 × 10^−4^
ChNS_1_GA, 15 °C	−0.50	2.17	41/173	0.99	0.37	4.46 × 10^−4^
ChNS_1_GA, 25 °C	−0.24	1.20	6/120	0.98	0.59	1.16 × 10^−3^
ChNS_1_GA, 35 °C	−0.08	0.83	~2/20	0.99	0.26	5.00 × 10^−4^
ChNS_1_GA, 45 °C	−0.05	0.77	~2/14	0.99	0.22	3.17 × 10^−4^

**Table 6 ijms-24-11818-t006:** The name, composition, and bulk density of the samples being tested.

Name of Sample	Composition	Bulk Density, d_b_, (g/dm^3^)
ChNS_05	A-300 – 25 g Chitosan (0.5 layer) – 1.5 g water – 18 g	260
ChNS_1	A-300 – 25 g Chitosan (1 layer) – 3 g water – 18 g	282
ChNS_1_GA	A-300 – 25 g Chitosan (1 layer) – 3 g water – 18 g 25% wt. solution of GA – 2 g	344
ChSG_05	C-35 – 25 g Chitosan (0.5 layer) – 1.5 g water – 18 g	290
ChSG_1	C-35 – 25 g Chitosan (1 layer) – 3 g water – 18 g	301
ChSG_1_GA	C-35 – 25 g Chitosan (1 layer) – 3 g water – 18 g 25% wt. solution of GA – 2 g	354

## Data Availability

The data are available by corresponding author.

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
