# Peer review of "Chitosan–Silica Composites for Adsorption Application in the Treatment of Water and Wastewater from Anionic Dyes"

_ijms, 2023, doi:10.3390/ijms241411818_

Round 1

Reviewer 1 Report

As it is nicely stressed in Introduction of the paper, Polysaccharide nanocomposites have become important materials over the last decade for various applications.  The biodegradability, biocompatibility, and non-toxicity features of polysaccharides make them an attractive and viable alternative to synthetic polymers especially for the biomedical uses. Among a wide group of biopolymers, chitosan has found a special place in modern scientific and technological approaches being the second most produced natural polymers. But chitosan is insoluble in water and in solutions with pH > 7, only in acidic conditions.

A big portion of the experimental work is done by authors. The introduction is wonderful. The manuscript is worth for publishing in the International Journal of Molecular Sciences after minor revision.

Comments

1.      The properties of materials are strongly dependent on the purity of the materials and purity of their surface. Though the materials were purchased from the commercial companies, how their quality were checked? Especially of chitosan. Properties, polymerization degree, mass, and purity of this polymer depends on producer.

2.      As far as I understand from the Figure 1 (also line 216) it follows that SAXS studies do not reveal visible difference in the composites and are mainly due to inorganic phase. But as Figure 4 shows there is a diffenrence in SEM images. Can the authors stress more precisely why SAXS do not reveal changes, I did not quite get that.

3.      What kind of mechanisms of the chitosan-silica cross-linking (bonding) are supposed to be exist in the obtained samples? Is it expected that the various ratio between the amounts of reagents and various preparation procedures, including pH, would result in different cross-linking? Can a radical mechanism, surface charging, etc, play a role in the cross-linking (bonding)? Which atoms are supposed to be responsible for the linkage? Can the authors give some brief review of the possible cross-linking ways in silica-chitosan (or other biopolymers) compounds and propose the best one which corresponds to the experimental findings?

4.      Did the authors investigate the thermal stability of composites?

Author Response

List of changes in the revised manuscript and detailed responses to Reviewers

Reviewer 1

Reviewer’s comment The properties of materials are strongly dependent on the purity of the materials and purity of their surface. Though the materials were purchased from the commercial companies, how their quality were checked? Especially of chitosan. Properties, polymerization degree, mass, and purity of this polymer depends on producer.

Our answer: We agree that chitosan as a material of natural origin may differ in properties and purity depending on the synthesis procedure and manufacturer. Purity and properties should be controlled. The additional information was added to the experimental part:

“Chitosan from shrimp shells with quality level QL=200 with molecular weight from 190 000 to 370 000 Da and a degree of deacetylation ~ 75%, was purchased from Sigma–Aldrich.”

Before the adsorption process, the chitosan-silica composite material as example of ChNS_1 was tested by XPS surface analysis for quality control. The analysis revealed a typical carrier-specific X-ray photoelectron spectrum and shifts in the C1s signal associated with chitosan. The resolution of the technique (the ability to analyze atoms with a content of more than 1%) did not allow to examine changes within amino groups and further analysis in this regard was discontinued. However, the presence of impurities and undesirable substances was not disclosed. Additional Figure (Figure 1_SI) was added to the Supplementary Indformation.

Reviewer’s comment As far as I understand from the Figure 1 (also line 216) it follows that SAXS studies do not reveal visible difference in the composites and are mainly due to inorganic phase. But as Figure 4 shows there is a diffenrence in SEM images. Can the authors stress more precisely why SAXS do not reveal changes, I did not quite get that.

Our answer: Figure 1 shows the experimental SAXS scattering curves, which, although very similar, are not identical. The greatest differences are revealed in the initial range of the scattering vector, which refers to nanometer objects. The similar course of the curves is related to the nature of the SAXS technique, which is a volumetric method. The largest amount of structural information is therefore related to the silica phase, which is in the majority. The changes observed in Fig. 4 concern the surface layer, which contributes little to the overall structural information. Here, information on interfacial properties should be extracted from the SAXS data, which are presented in Fig. 2 or Fig. 3, which show the size distribution of inhomogeneities, which also clearly differentiates the tested materials.

Reviewer’s comment What kind of mechanisms of the chitosan-silica cross-linking (bonding) are supposed to be exist in the obtained samples? Is it expected that the various ratio between the amounts of reagents and various preparation procedures, including pH, would result in different cross-linking? Can a radical mechanism, surface charging, etc, play a role in the cross-linking (bonding)? Which atoms are supposed to be responsible for the linkage? Can the authors give some brief review of the possible cross-linking ways in silica-chitosan (or other biopolymers) compounds and propose the best one which corresponds to the experimental findings?

Our answer: Additional explanation was added to the manuscript:

To explain the specificity of forming a chemically resistant composite, a short description of the cross-linking method using various cross-linking agents is here presented. Generally, the idea of cross-linking is the formation of a polymer network based on chemical or physical bonds (covalent and/or ionic) between polymer chains or between polymer chains and a multifunctional cross-linking agent. Depending on the type of cross-linking agent used, cross-linking takes place with the participation of amino group or hydroxyl group at the C-6 atom of the polymer chain. In the case of using glutaraldehyde (GA) or genipin (G), the process takes place between chitosan amino groups and aldehyde groups of GA (covalent imine bond) or ester groups of G (covalent amine bond), respectively. If epichlorohydrin (ECH) is used, the hydroxyl group of chitosan at the C-6 atom and a reactive oxirane ring of ECH are involved in the process. Another way of chitosan cross-linking is the addition of negatively charged molecules (complexing inorganic ions, sulfuric acid, and etc.) or macromolecules (alginate, pectin, hyaluronic acid, and etc.). A type of cross-linker and its concentration, an average molecular weight of chitosan and its deacetylation degree, a concentration of chitosan solution and a cross-linking time affect a final product density. Chitosan can also undergo cross-linking under influence of electron beam, gamma radiation, UV radiation, enzymes, or heating [64-67]. The specific conditions in the ball mill during the synthesis of chitosan-silica composite (ChNS) suggest that thermally-induced crosslinking of chitosan took place, providing chemical stability of the solid during the adsorption process in a strongly acidic solution. Strong aggregation and agglomeration of nanosilica particles containing  groups can be treated here as a polyanionic substance, and chitosan as a polycationic one. On the other hand formation of covalent bonds (Si−O−С) between polymer carbinol groups and silica silanol groups can be expected. Thus, considered cross-linking process is based on a thermally-induced mechanism of ionic and/or covalent interactions.”

Reviewer’s comment Did the authors investigate the thermal stability of composites?

Our answer: Due to the extensive size of the work submitted the authors are planning to devote a separate article on the thermal stability of obtained chitosan-silica composites based on the thermal analysis measurements (thermogravimetry, TG; derivative thermogravimetry, DTG; and differential scanning calorimetry, DSC) coupled with FTIR and MS spectrometers. Undoubtedly it will be a valuable supplement to the results presented here.

Reviewer 2 Report

The significante of polysaccharide nanocomposites as materials in the context of both soft and hard nanomaterials cannot be overstated. This  manuscript provides an insightful analysis into the various structural, textural, and physicochemical aspects of chitosan-silica adsorbents, employing a range of methodologies.

This study serves as a valuable resource, offering novel insights specifically within the field of nanocomposites. The investigative methods detailed are comprehensive and impressive.

My primary critique is the manuscript's length. A more concise version, preserving key details, would improve readability and comprehension.

Should these revisions be successfully impleented, I would recommend this paper for publication.

Author Response

List of changes in the revised manuscript and detailed responses to Reviewers

Reviewer 2

Reviewer’s comment: My primary critique is the manuscript's length. A more concise version, preserving key details, would improve readability and comprehension.

Our answer: As directed, the manuscript has been redrafted into a more concise form. Especially:

  • Figure 5 has been moved to Supplementary Information as Figure 3_SI
  • Figure 7 was removed.
  • Figure 12 has been moved to Supplementary Information as Figure 4_SI
  • Figure 18 has been moved to Supplementary Information as Figure 5_SI
  • Table 6 has been moved to Supplementary Information as Table 1_SI

Reviewer 3 Report

There are too many figures, I suggest uniting some of them in one.

 Minor editing of English language required.

Author Response

List of changes in the revised manuscript and detailed responses to Reviewers

Reviewer 3

Reviewer’s comment: Abstract just summarizes the performed experiments and no results are indicated. Please, include some data and relevant results.

Our answer: The Abstract part was reworded according to suggestion. The following sentences were added to the Abstract section:

“The highest adsorption efficiency was achieved at pH=2, hence the desirability of modifications aimed at stabilizing chitosan in such conditions. The increase of chitosan amount in the synthesis grows 1.8 times the adsorption capacity for nanosilica-supported material and 1.6 for silica gel-based composites. The adsorption kinetics of anionic dyes (acid red AR88) was faster for ChNS than for ChSG which results from silica-type effect.”

Reviewer’s comment: Some references are missing in the introduction: - Pag 2, line 52: The importance of chitosan can be illustrated by the fact that it is expected to rank second after cellulose in the group of mass-produced natural polymers. Please, provide a reference or specific data.

Our answer: The missing references were added:

  1. Rinaudo, M. Chitin and chitosan: Properties and applications. Progress in Polymer Science 2006, 31, 603-632, doi:10.1016/j.progpolymsci.2006.06.001.
  2. Mishra, R.; Militky, J. Nature, nanoscience, and textile structures. Nanotechnology in Textiles: Theory and Application 2019, 1-34, doi:10.1016/b978-0-08-102609-0.00001-8.

Reviewer’s comment: The whole introduction is focused on chitosan-based materials and their applications. However, any silica composite research is mentioned. I suggest the authors reducing the chitosan explanations that are not related to the application presented in this work and including some silica references with a short explanation.

Our answer: The Introduction part was reworded according to the suggestion:

The part of the text was removed:

“For biomedical applications magnetic systems for specialized drug delivery systems were also proposed. In the case of targeted therapy, it is important that the drug reaches its destination safely, where it will be detached from the carrier. Research reports in this area concern magnetic chitosan/hydroxyapatite nanoparticle as a protein drug carrier [58], novel nanospheres for controlled release of the anticancer drug, 5-fluorouracil (5-FU) [59] and magnetic microspheres prepared by microemulsion polymerization for protein drug delivery systems [58].”

The additional part was added:

“Despite such versatile examples, the applications related to the use of chitosan are still subject to certain limitations. In many applications, amplification and modification of complex systems is required. The solution may be related to the creation of composite systems, e.g. silica or carbon materials [58-61].”

Reviewer’s comment: Why have you chosen silica for these nanocomposites? This question should be addressed along the introduction. The purpose of using chitosan is clear.

Our answer: The text of the Introduction was amended by adding (in bold):

“It has been assumed that a chitosan-silica composite would work more effectively as an adsorbent due to the presence of the chitosan component. It is well known that due to its polycationic nature, chitosan is characterized by remarkable sorption properties towards anionic pollutants. However, it is a substance with low mechanical, thermal, and chemical stability. Therefore, the combination of chitosan with silica (which role is to be a rigid support for polymer due to its high stability) was aimed at obtaining a composite that would exhibit both satisfactory sorption properties and mechanical and thermal stability. To improve the chemical stability of the solid, during synthesis, it was subjected to the cross-linking process with glutaraldehyde (GA). The choice of silica as inorganic matter was also dictated by the presence of the silanol groups (Si–OH) on its surface, enabling immobilization of organic matter.”

Reviewer’s comment: Which is the resolution of the X-rays scattering in the range of small diffraction angle

experiments?

Our answer: SAXS patterns are collected at small angles of a few degrees (0.2-5 2 theta). SAXS technique is capable of delivering structural information in the resolution range between 1 and 100 nm, and even up to 150 nm for distances in partially ordered systems. A special type of this technique Ultra small-angle scattering (USAS) can resolve even larger dimensions.

Reviewer’s comment: The meaning of each abbreviation should be explained the first time it appears. I have

detected several issues regarding this.

Our answer: It was corrected

Reviewer’s comment: Line 168 should it be Table 1 instead of 2?

Our answer: It was corrected

Reviewer’s comment:  Line 368 check “(ChSG_1 and ChSG_1)”

Our answer: It was corrected

Reviewer’s comment: Part of section “2.2. Adsorption properties of the chitosan-silica composites” is the missing

part in the introduction. I suggest the authors moving these paragraphs to the intro.

Our answer: It was corrected

Reviewer’s comment: The description of the composites’ synthesis should be in the first place, before any

experiment in order to help the reader. I suggest moving section 2.2 before 2.1

Our answer: Thank you for the tip. At this point, changing the order involves changing the numbering of figures, tables and literature, so to avoid internal errors, we prefer that the order remain in this form. However, we will try to take this opinion into account in the future.

Reviewer’s comment:Line 499, the authors say “Moreover, adsorption takes place in interparticle spaces (voids) and not in voids inside the material (pores). The latter, due to their inappropriate size or shape, may limit or exclude adsorption.” Why? There are many publications in which monolithic silica aerogels are used for adsorption processes through their pores. Please, check this statement.

Our answer: We thank Reviewer for the valuable remark, because in the case of the composites based on SG, there is no exclusion effect towards adsorbate (share of micropores in the total porosity is negligible). So, the indicated sentence was changed as follows:

“Moreover, adsorption takes place in interparticle spaces (voids) of the material. and not in voids inside of the material (pores). The latter, due to their inappropriate size or shape, may limit or exclude adsorption.”

Reviewer’s comment: Any picture of the final composites is included in the manuscript, it would be highly recommended to include them. I understand that you obtain nanoparticles, so after the adsorption process, a filtration step would be necessary? It would reduce the efficiency of the process.

Our answer: The missing SEM figures (Figure 4g and 4h) were added.

Our answer: Before spectrophotometric measurements, solutions after dye adsorption process on composites were separated using a high-speed centrifuge. In our opinion, based on earlier studies, the applied separation technique does not affect the process efficiency. In the case of the kinetic measurements, it was necessary to make filters that would prevent from getting an adsorbent with a solution into a measuring cell. For this purpose, a 2 ml syringe, glass wool, and mesh nylon material were used. The handmade filter is presented in the photo below:

Reviewer’s comment: Have the authors studied the reusability of the composites? It is a relevant aspect.

Our answer: The synthesized composites have not been tested on the regeneration and reusability abilities but in authors’ opinion and after the specific bibliography analysis one can assume that the materials would get a positive result. In the subsequent research studies the authors will take advantage of the valuable reviewer’s suggestion for which thank you very much.

Reviewer’s comment: Authors should include some dye adsorption experiments with the pure NS, SG and chitosan in order to confirm the synergistic effect.

Our answer: The dye adsorption experiments with the pure NS, and chitosan were conducted and their results were compared with that on the composite ChNS_1.

“In Figure 11b the kinetic profiles for AR88 adsorption on the pure chitosan (Ch), nanosilica (NS) and nanosilica-based composite (ChNS_1) are presented. Applied weights of Ch and NS corresponded to their amounts in ChNS_1 calculated based on the synthesis procedure. A much faster adsorption on the composite compared to that on the chitosan is observed, while adsorption on the nanosilica did not occur. It proves the synergistic effect of the presence of dye-binding groups of chitosan component and the developed interfacial surface of nanosilica component in the composite.”

Figure 11. Kinetic profiles for AR88 adsorption on the pure chitosan (Ch), nanosilica (NS) and nanosilica-based composite (ChNS_1) (b).

Reviewer’s comment: Additionally, authors could study the presence of covalent bonds between matrixes by FTIR.

Our answer: Discussion of FTIR measurements have been added to Supplementary Information.

“In order to confirm the SiO2 and chitosan interaction (covalent bonds) FTIR spectra were obtained for the initial chitosan and synthesized composite (ChNS_1). In the FTIR spectrum of chitosan the band at 3440 cm−1 corresponds to the stretching vibrations of hydroxyl groups O-H. Intensive absorption bands at 2880 to 2890 cm−1 are observed due to the С-Н stretching vibrations. Position 1565 cm−1 corresponds to the deformation vibra-tions of -NH2 and 1420 сm−1 for C-H bending vibrations. Asymmetric С-О-С stretching vibrations were found at 1300 сm−1 and 1070 сm−1 for С-О stretching vibration of СН-ОН. In the case of the composite, we were looking for the vibration shift of the groups potentially involved in such bonding. These groups were the -NH2 forms. The shift was observed in the range of 1565-1550 cm-1 and is shown in Figure 6_SI. The content of chitosan in the composite was very low, therefore the intensity is relatively low, although observable.”

Figure 6_SI. Structure characterization of chitosan and chitosan-silica nanocomposite (ChNS_1) by FTIR spectra. As insets: the chemical structure of chitosan and their functional groups correspond to the representative bands on the FTIR spectra and shift of position characteristic for –NH2 groups as evidence of their interaction in composite system.

Reviewer’s comment: Finally, the authors say that the composites were dried at 40 ˚Cc for 5 h. Do the particles experiment a shrinkage? This is a relevant aspect that should be studied

Our answer: The process of drying samples is one of the stages of the synthesis of materials, and only in this form they were used for the adsorption process. No additional effects related to further drying of the final product were noticed.

Reviewer 4 Report

This work reports the fabrication of chitosan-silica and chitosan-silica gel nanocomposites by mechanical treatment. Moreover, the adsorption properties of the prepared materials are comprehensively investigated. Overall, the amount of results is impressive, and the related discussions are sensible. Only a few minor points are required to be improved prior to publication in IJMS.

1. The article title is too general since there are only the chitosan-silica and chitosan-silica gel composite systems studied. The authors are recommended to change the title to be more specific.

2. The authors should provide more information in the introduction section on why acid red 88 is selected to use in this research.

3. The elemental analysis is not described in the methodology section. The authors should add some texts describing the method of elemental microanalysis in the SEM paragraph.

4. The bulk analysis for elemental compositions (e.g., XRF measurement) is recommended to be studied to support the results from EDS.

5. SEM images for all the samples are recommended to be added to the recent Figure 4.

6. Nitrogen adsorption isotherms and their corresponding BJH plots for all the samples are suggested to be added to the main text or supporting information to support the data in Table 1.   

7. The authors should provide the thermal analyses of the samples to confirm the crosslinking in the samples with GA added.

8. The authors are recommended to include more insightful details regarding the adsorption mechanism.

Author Response

List of changes in the revised manuscript and detailed responses to Reviewers

Reviewer 4

Reviewer’s comment: The article title is too general since there are only the chitosan-silica and chitosan-silica gel composite systems studied. The authors are recommended to change the title to be more specific.

Our answer: The title has been clarified following the reviewer's comments:

“Chitosan-silica composites for adsorption application in the treatment of water and wastewater from anionic dyes.”

Reviewer’s comment: The authors should provide more information in the introduction section on why acid red 88 is selected to use in this research.

Our answer: The following sentences were added to the Introduction  section:

“In this work the structural modifications of chitosan-silica materials and their impact on the efficiency of the adsorption process of anionic colouring substances is examined. As a model adsorbate Acid red 88 (AR 88) belonging to the sulfonic azo substances was chosen. Due to the widespread usage of this group of dyes in the textile, cellulose, paper, chemical, food, and cosmetic industries, they are a main component of industrial wastewater and a dangerous source of environmental pollution. On the other hand, the complex structure of dyes’ molecules makes them highly stable to light and oxidizing agents as well as resistant to biodegradation. Thus, the usefulness of some traditional methods in removing dyes from aqueous solutions is limited. The effectiveness of flocculation and coagulation processes with metal compounds in water decoloration is high, but the consequence of their usage is the formation of sludge and an increase in the concentration of metallic pollutants in water. Therefore, the adsorption techniques are a good alternative to conventional dye removal methods.”

Reviewer’s comment: The elemental analysis is not described in the methodology section. The authors should add some texts describing the method of elemental microanalysis in the SEM paragraph.

Reviewer’s comment: The bulk analysis for elemental compositions (e.g., XRF measurement) is recommended to be studied to support the results from EDS.

Our answer: We will explain these two issues together because both questions are related. The elemental analysis presented in Table 2 was performed by organic elemental analysis using CHNS/O Analyzer. Thus, the applied technique is based on determining carbon (C), hydrogen (H), and nitrogen (N), contents in investigated samples by the complete combustion of the sample for detection of even small elemental quantities in bulk nature.

The following sentences were added to section 3.3.2. Other technique:

“Carbon, hydrogen, and nitrogen analysis of the chitosan–silica composites was carried out by using Series II CHNS/O Analyzer 2400 (Perkin Elmer, USA). The temperature of the reduction and the combustion processes were 650 and 950°С, respectively. The 500 mg of each sample was used during analysis.”

Reviewer’s comment: SEM images for all the samples are recommended to be added to the recent Figure 4.

Our answer: In the presented version of the manuscript, we focused only on the observation of morphological changes for the most diverse samples, i.e. ChNS_1 ChNS_1_GA, and ChSG_1. According to the Reviewer's comments, SEM images for the ChSG_1_GA sample (Fig. 4g and 4h) have been added to Figure 4.

Reviewer’s comment: Nitrogen adsorption isotherms and their corresponding BJH plots for all the samples are suggested to be added to the main text or supporting information to support the data in Table 1.

Our answer: Nitrogen adsorption isotherms and their corresponding BJH plots were added to the Supplementary Information (Fig. 2_SI).

Reviewer’s comment: The authors should provide the thermal analyses of the samples to confirm the crosslinking in the samples with GA added.

Our answer: Due to the extensive size of the work submitted the authors are planning to devote a separate article on the thermal stability of obtained chitosan-silica composites based on the thermal analysis measurements (thermogravimetry, TG; derivative thermogravimetry, DTG; and differential scanning calorimetry, DSC) coupled with FTIR and MS spectrometers. Undoubtedly it will be a valuable supplement to the results presented here.

Reviewer’s comment: The authors are recommended to include more insightful details regarding the adsorption mechanism.

Our answer: Additional explanation was added to the manuscript in discussion of adsorption experiments.

“The adsorption process in the discussed systems was carried out under neutral pH conditions, therefore the mechanism of binding the dye to the composite was based on attractive electrostatic interactions between partially protonated amino groups of chitosan and dye anions, as well as hydrogen bonds involving functional groups of the composite (i.e. amino, amide, hydroxyl, silanol ) and dye (azo, sulfonic, hydroxyl functional groups). Depending on adsorption conditions, especially ionic strength and pH level, the mechanism of the process can be changed including diversification of attractive/repulsive interactions, strength of hydrogen bonding, partial share of complexation.”

More detailed considerations regarding the adsorption mechanism are included in the subsection 2.2.4. Effect of pH on the adsorption process.

Round 2

Reviewer 3 Report

Authors successfully answered the reviewer's comments and performed additional experiments.

Congratulations for your work.